# Linking spatial self-organization to community assembly and biodiversity

**Bidesh K Bera[1], Omer Tzuk[2†], Jamie JR Bennett[1], Ehud Meron[1,2]***

[1]Department of Solar Energy and Environmental Physics, BIDR, Ben-Gurion University of the Negev, Sede Boqer Campus, Israel; [2]Physics Department, Ben-Gurion University of the Negev, Beer Sheva, Israel

**Abstract** Temporal shifts to drier climates impose environmental stresses on plant communities that may result in community reassembly and threatened ecosystem services, but also may trigger self-organization in spatial patterns of biota and resources, which act to relax these stresses. The complex relationships between these counteracting processes – community reassembly and spatial self-organization – have hardly been studied. Using a spatio-temporal model of dryland plant communities and a trait-based approach, we study the response of such communities to increasing water-deficit stress. We first show that spatial patterning acts to reverse shifts from fast-growing species to stress-tolerant species, as well as to reverse functional-diversity loss. We then show that spatial self-organization buffers the impact of further stress on community structure. Finally, we identify multistability ranges of uniform and patterned community states and use them to propose forms of non-uniform ecosystem management that integrate the need for provisioning ecosystem services with the need to preserve community structure.

**\*For correspondence:**
ehud@bgu.ac.il

**Present address:** [†]Department of Industrial Engineering, Faculty of Engineering, Tel-Aviv University, Tel Aviv-Yafo, Israel

**Competing interests:** The authors declare that no competing interests exist.

## Introduction

The structure of plant communities – their composition and diversity – forms the foundation of many ecosystem services on which human well-being crucially depends. These include provisioning services such as food, fresh water, wood, and fiber; regulating services such as flood regulation and water purification; cultural services such as recreation and aesthetic enjoyment; and supporting services such as soil formation, photosynthesis, and nutrient cycling (*Duraiappah and Naeem, 2005*). These services are at risk due to potential changes in the composition and diversity of plant communities as a result of global warming and the development of drier climates (*Harrison et al., 2020*). Understanding the factors that affect community structure in varying environments calls for integrated studies of mechanisms operating at different levels of organization, from phenotypic changes at the organism level, through *intra*specific interactions at the population level, to *inter*specific interactions at the community level (*Gratani, 2014*; *Falik et al., 2003*; *Bertness and Callaway, 1994*; *Pérez-Ramos et al., 2019*). Of these mechanisms, the role of intraspecific interactions in driving community dynamics through spatial self-organization, has hardly been studied (see however *Vandermeer and Yitbarek, 2012*; *Bonanomi et al., 2014*; *Cornacchia et al., 2018*; *Zhao et al., 2019*; *O'Sullivan et al., 2019*; *Pescador et al., 2020*; *Inderjit et al., 2021*).

Spatial self-organization in periodic and non-periodic patterns of biota and resources, driven by intraspecific competition that leads to partial plant mortality, is widely observed in stressed environments (*Rietkerk and van de Koppel, 2008*). An important class of these phenomena are vegetation patterns in drylands. In sloped terrains, these patterns commonly appear as parallel vegetation stripes oriented perpendicular to the slope direction (*Lefever and Lejeune, 1997*; *Valentin et al., 1999*; *Bastiaansen et al., 2018*). In flat terrains, with no preferred direction imposed by slope or wind, stripe-like configurations often appear as labyrinthine patterns. However, in such terrains two additional pattern morphologies are often observed; nearly periodic patterns of bare-soil gaps in

otherwise uniform vegetation, and nearly periodic patterns of vegetation spots in otherwise uniform bare soil (*Rietkerk et al., 2004*; *Deblauwe et al., 2008*; *Borgogno et al., 2009*; *Getzin et al., 2016*). Spatial self-organization may not necessarily result in periodic patterns; according to pattern-formation theory (*Meron, 2015*; *Knobloch, 2015*), it can also result in non-periodic patterns, such as single or randomly scattered vegetation spots in otherwise bare soil, randomly scattered bare-soil gaps in otherwise uniform vegetation and others (*Tlidi et al., 2008*; *Dawes and Williams, 2016*; *Parra-Rivas and Fernandez-Oto, 2020*; *Jaïbi et al., 2020*; *Zelnik et al., 2015*). The driving mechanisms of these self-organized vegetation patterns are scale-dependent positive feedback loops between local vegetation growth and water transport toward the growth location (*Rietkerk and van de Koppel, 2008*; *Pringle and Tarnita, 2017*; *Meron, 2019*).

Vegetation patterns involve not only spatial distributions of plant biomass, but also less-visible distributions of soil-water, nutrients, soil biota, and possibly toxic substances (*Paz-Kagan et al., 2019*; *Inderjit and Duke, 2003*; *De Deyn et al., 2004*; *van der Putten et al., 2013*). The various habitats that these self-organizing distributions form lead to niche differentiation and community reassembly (*Weiher et al., 2011*). Thus, spatial self-organization and community dynamics are intimately coupled processes that control community composition and diversity. Understanding this unexplored coupling is essential for assessing the impact of global warming and drier climates on community structure and ecosystem services.

In this paper, we incorporate spatial self-organization into community-assembly studies, using a mathematical model of dryland plant communities. Our model study provides three new insights, illustrated in *Figure 1*: (i) Spatial self-organization acts to reverse community-structure changes induced by environmental stress, (ii) it buffers the impact of further stress, and (iii) it offers new directions of ecosystem management that integrate the need for provisioning ecosystem services with the need to conserve community structure. More specifically, using a trait-based approach (*Díaz and Cabido, 2001*), we show that drier climates shift the composition of spatially uniform communities toward stress-tolerant species, and reduce functional diversity, that is the diversity of functional traits around the most abundant one. By contrast, self-organization in spatial patterns, triggered by these droughts, shifts the composition back to less tolerant species that favor investment in growth, and increase functional diversity. Once patterns are formed, spatial self-organization provides various pathways to relax further stresses without significant changes in community composition and diversity. Furthermore, multistability ranges of uniform and patterned states open up new opportunities for grazing and foddering management by forming mixed community states of increased functional diversity.

## Results

### Modeling spatial assembly of dryland plant communities

We refer the reader to the Materials and methods section for a detailed mathematical presentation of the model we use in this study, and restrict the description here to a few essential aspects and properties. The model largely builds from foundations introduced earlier (*Meron, 2016*). These foundations capture three pattern-formation mechanisms associated with three forms of water transport: overland water flow, soil-water diffusion, and water conduction by laterally spread roots (*Meron, 2019*). In this study we focus, for simplicity, on a single mechanism associated with overland water flow. In practice, this may be the case with plant species that have laterally confined roots and compact soils where soil-water diffusion rates are low. The mechanism induces a Turing instability of uniform vegetation, leading to periodic vegetation patterns. The instability can be understood in terms of a positive feedback loop between local vegetation growth and overland water flow toward the growth location: an area with an incidental denser vegetation draws more water from its surrounding areas than the latter do, which accelerates the growth in that area and decelerates the growth in the surrounding areas, thus amplifying the initial nonuniform perturbation. This feedback loop is also called a scale-dependent feedback because of the positive effect on vegetation growth at short distances and negative effect at longer distances (*Rietkerk and van de Koppel, 2008*). The reason why an area of denser vegetation draws more water from its surroundings is rooted in the differential infiltration of overland water into the soil that develops – high in denser vegetation and low in sparser vegetation, as illustrated in *Figure 2*. Several factors contribute to this process, including

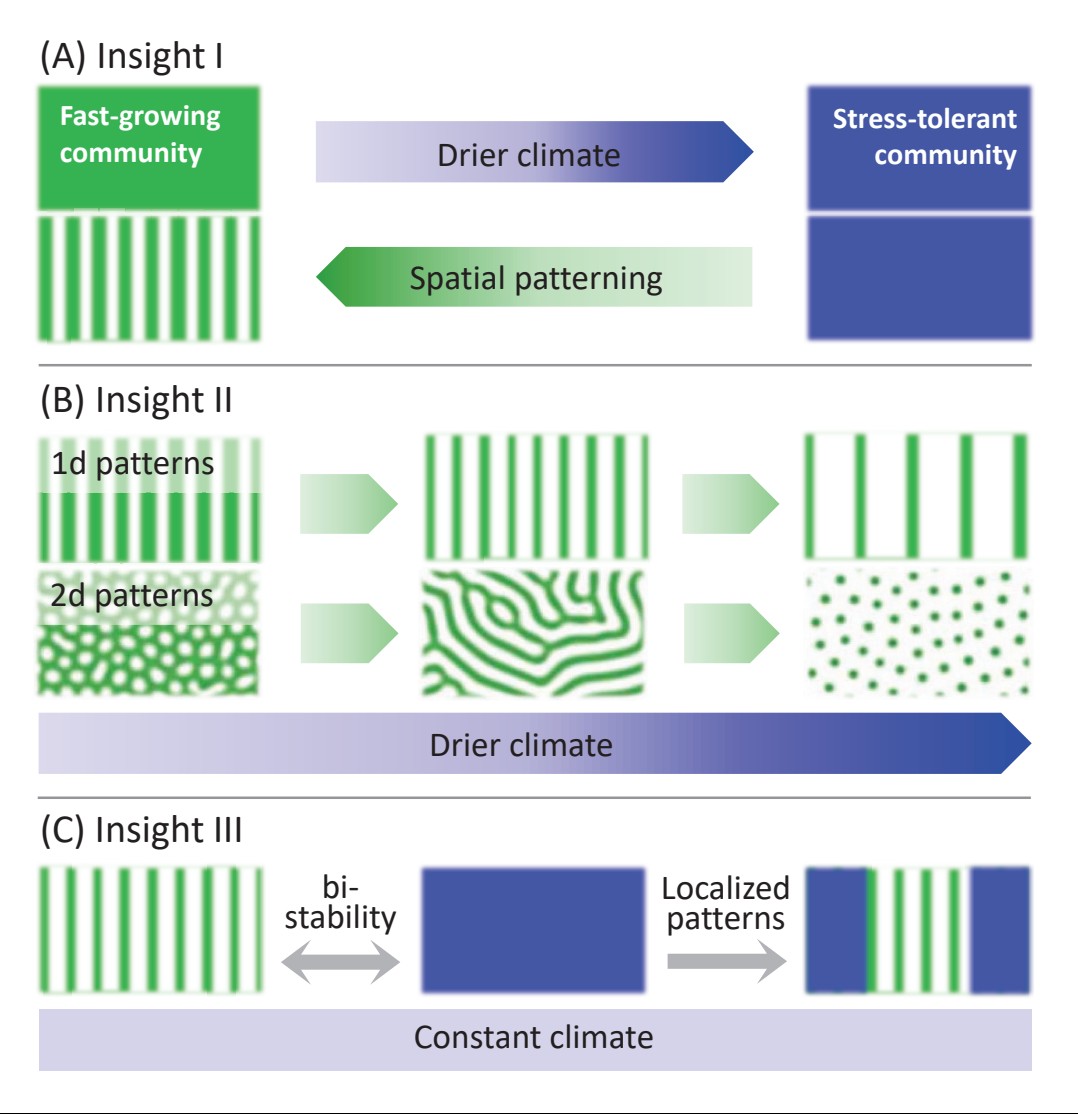

**Figure 1.** A schematic illustration of the three insights that the model study provides. (**A**) Insight I: A drier climate shifts an original spatially uniform community of fast-growing plants, denoted by a green color, to a uniform community of stress-tolerant plants, denoted by blue color. Spatial patterning induced by the drier climate shifts the community back to fast growing plants. (**B**) Insight II: Once patterns have formed a yet drier climate has little effect on community structure – all patterned community states consist of fast-growing plants (green). This is because of further processes of spatial self-organization that increase the proportion of water-contributing bare-soil areas and compensate for the reduced precipitation. In one-dimensional (1d) patterns, these processes involve thinning of vegetation patches or transitions to longer wavelength patterns. In two-dimensional (2d) patterns, these processes involve morphological transitions from gap to stripe patterns and from stripe to spot patterns. (**C**) Insight III: Localized patterns in a bistability range of uniform and patterned community states significantly increase functional diversity, as they consist of both stress-tolerant (blue) and fast-growing (green) species. Such patterns can be formed by nonuniform biomass removal as an integral part of a provisioning ecosystem service.

denser roots in denser-vegetation areas, which make the soil more porous, and lower coverage of the ground surface in denser-vegetation areas by physical or biological soil crusts that act to reduce the infiltration rate. The differential infiltration induces overland water flow toward areas of denser vegetation that act as sinks.

We describe the plant community using a trait-based approach that shifts the focus from species, and the many traits by which they differ from one another, to groups of species that share the same values of selected functional traits. Since the general context is ecosystem response to drier

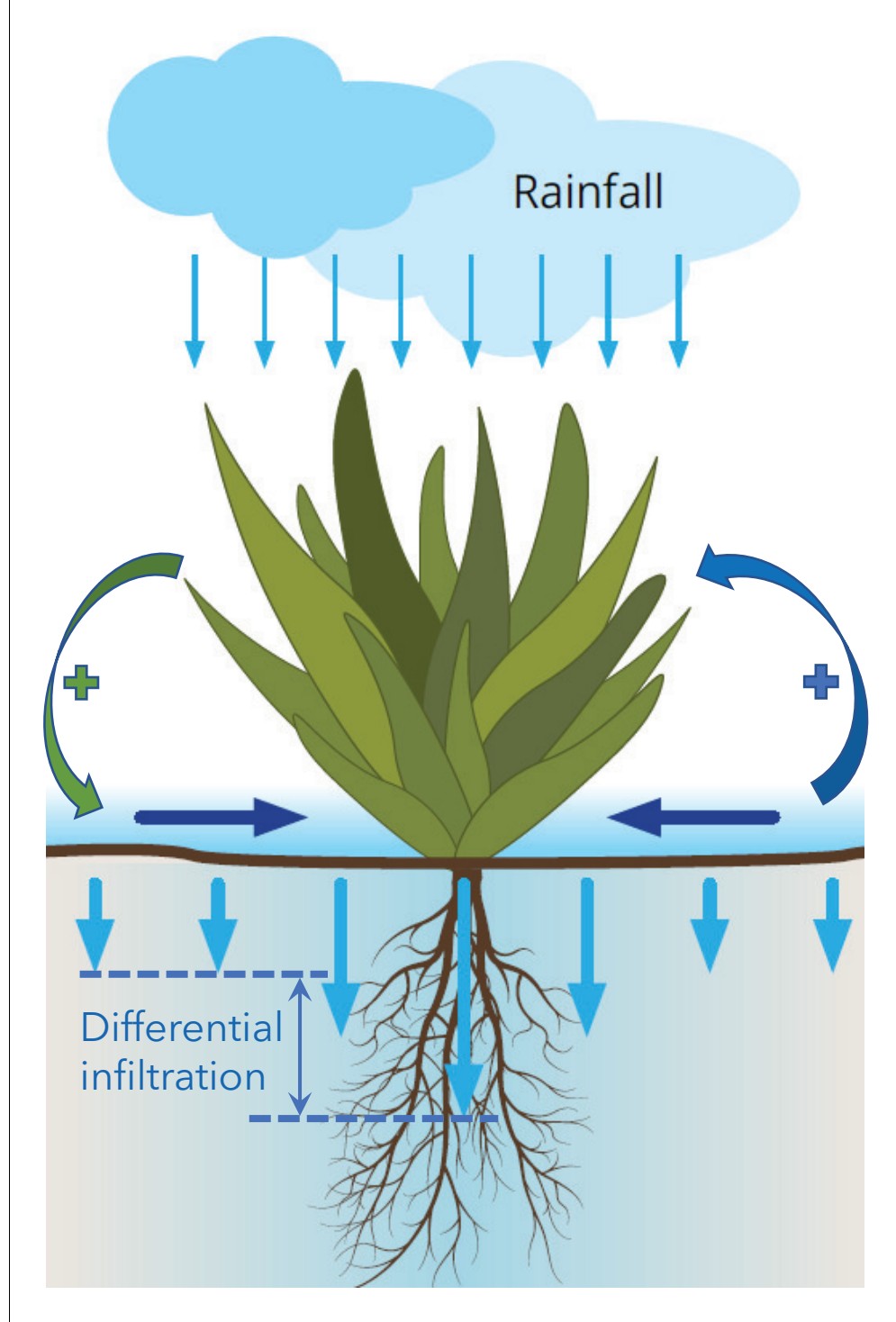

**Figure 2.** Illustration of overland water flow toward vegetation patches (horizontal arrows), induced by differential infiltration: low in bare soil (short vertical arrows) and high in vegetation patches (long arrows). Vegetation growth enhances the infiltration contrast and thus the overland water flow (green round arrow), while that flow further accelerates vegetation growth (blue round arrow). The two processes form a positive feedback loop that destabilizes uniform vegetation to form vegetation patterns, and acts to stabilize these patterns once formed.

climates, we choose the functional traits to include a response trait associated with tolerance to water stress and an effect trait associated with shoot growth and biomass production (*Suding et al., 2008*). We further assume a tradeoff between the two traits which is well supported by earlier studies (*Angert et al., 2009*; *Dovrat et al., 2019*). Thus, we distinguish between different functional groups through the different tradeoffs they make between shoot growth and tolerance to water stress. The plant community is quantified by introducing a dimensionless trait parameter (*Nathan et al., 2016*; *Tzuk et al., 2019*; *Yizhaq et al., 2020*), $\chi \in (0, 1)$, that represents a tradeoff between plant investment in shoot growth vs. investment in tolerance to water stress, so that $\chi \to 0$ represents plants that invest mostly in growth, while $\chi \to 1$ represents plants that invest mostly in tolerating water stress (see the Materials and methods section for a mathematical formulation). Using this parameter, the pool of species is divided into $N \gg 1$ functional groups, where all species within the $i$ th group have χ values in the small interval $\Delta\chi = 1/N$ that precedes $\chi_i = i\Delta\chi$.

We quantify each functional group $i$ by a continuous biomass variable $B_i$ that represents the total above-ground biomass per unit area of plant species belonging to the functional group, irrespective of the number of individuals that contribute to this biomass density (*Meron et al., 2019*). Accordingly, we describe dominant functional groups as 'abundant' or 'productive' interchangeably. Since we consider large $N$ values we may regard the set of biomass variables $B_1, B_2, ..., B_N$ as a discretized form of a continuous function $B = B(\chi)$, where $B_i = B(\chi_i)$. The reader is referred to the Materials and methods section for a brief discussion of the limit $N \to \infty$ at which the biomass becomes a continuous function of χ.

We assume that all functional groups are present in the system, even if at diminishingly small biomass densities, which may represent long-lived seeds (*DeMalach et al., 2021*). Community assembly then proceeds on ecological time scales set by interspecific competition for water and light (see Materials and methods section). However, late competition stages involving similar traits, as dominant species out-compete others (*Tilman, 1982*), may proceed on much longer time scales, as *Figure 3a* shows. This is because of self-shading (*Noy-Meir, 1975*) that slows down the late growth stage of the most competitive species. We therefore need to consider another important process occurring on these long time scales, namely, mutations to species belonging to nearby functional groups. As *Figure 3b* shows, the effect of mutations, modelled by trait diffusion (see Materials and methods section), results in an asymptotic biomass distribution of finite width that represents the emerging community. That distribution contains information about several community-level properties, including functional diversity, community composition, and total biomass (area under the distribution curve). Functional diversity can be characterized by several metrics (*Mason et al., 2005*), including functional richness *FR* and evenness *FE* (see Materials and methods section). Community composition can be characterized by the trait $\chi_{max}$ of the most abundant or productive functional group, in conjunction with *FR* and *FE*. All three metrics can be easily derived from the biomass distribution, as *Figure 3b* illustrates (see also Materials and methods section). In the following we will describe community composition changes by referring mostly to $\chi_{max}$, as the most indicative metric of community shifts from fast-growing to stress-tolerant species and vice versa. We wish to point out that including additional growth-inhibiting factors, modeling e.g. the effect of pathogens, can lead to asymptotic biomass distributions of finite width without resorting to trait diffusion on evolutionary time scales.

## Single functional-group states

It is instructive to consider first solutions of a model for a single functional group. *Figure 4a* shows a bifurcation diagram calculated for $\chi = 1$, the functional group of species investing mostly in tolerance to water stress. A uniform vegetation solution (*UV*) exists and is stable at sufficiently high precipitation (*P*), but loses stability in a subcritical Turing bifurcation (*Meron, 2018*) as *P* is decreased below a threshold $P_T(\chi)$. That instability creates a bistability range of uniform vegetation and periodic patterns (*PP*) where hybrid states (*HS*), consisting of patterned domains of increasing size in otherwise uniform vegetation, exist (*Knobloch, 2015*; *Zelnik et al., 2015*). Besides the periodic-pattern solution that appears at the Turing bifurcation ($PP_{WL=81}$), many more periodic solutions appear, as *P* further decreases, with longer wavelengths (WL) (*Zelnik et al., 2013*; *Siteur et al., 2014*). The second periodic solution shown in the diagram ($PP_{WL=150}$) has a wavelength almost twice as long. A solution describing bare soil (*BS*), devoid of vegetation, exists at all *P* values but is stable only below a threshold value $P_B$. Similar bifurcation diagrams are obtained for lower χ values, representing faster-

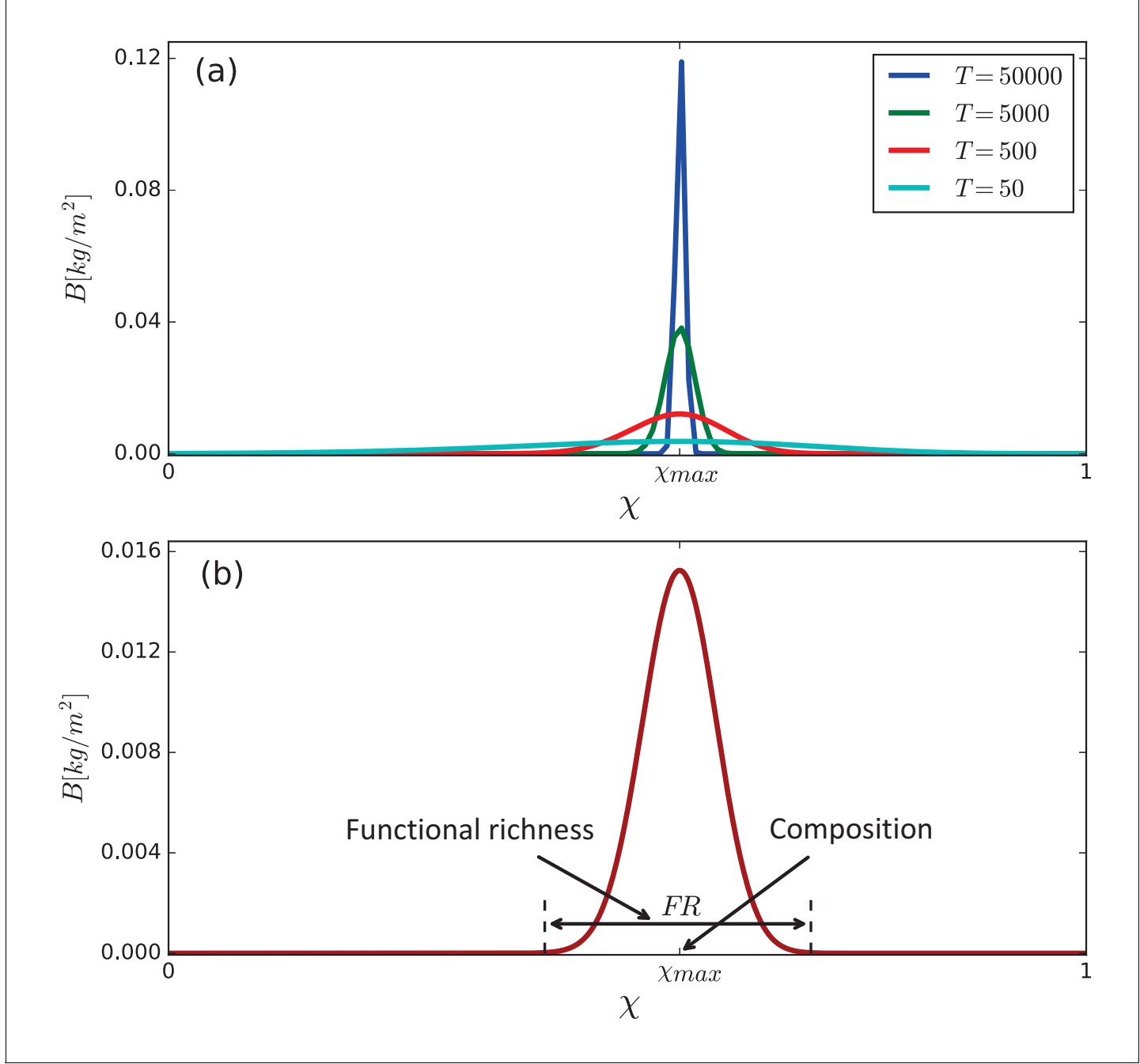

**Figure 3.** Emergence of a community as a stationary solution of the model *Equations (1)*. (a) Competitive exclusion of species in the course of time $T$[y] when trait diffusion is not allowed ($D_\chi = 0$). (b) Asymptotic biomass distribution obtained with slow trait diffusion ($D_\chi = 10^{-6}$). The distribution contains information about community-level properties such as community composition, quantified, among other metrics, by the position, $\chi_{max}$, of the most abundant functional group, and functional richness, quantified, among other metrics, by the distribution width, $FR$, at small biomass values representing the biomass density of a seedling. Parameter values: $P = 180$ mm/y and as stated in *Table 1*.

growing species, but the existence and stability ranges of the various solutions change. As *Figure 4b* shows, the uniform-vegetation state of species investing mostly in growth (low $\chi$) lose stability to periodic patterns at higher $P$ values. Also, the bare-soil state remains stable at higher $P$ values. These results imply that species investing in growth have a stronger tendency to form patchy vegetation. This is because of the steeper gradient of infiltration rates that faster-growing species create, which enhances overland water flow and, thereby, facilitates the Turing instability. The results

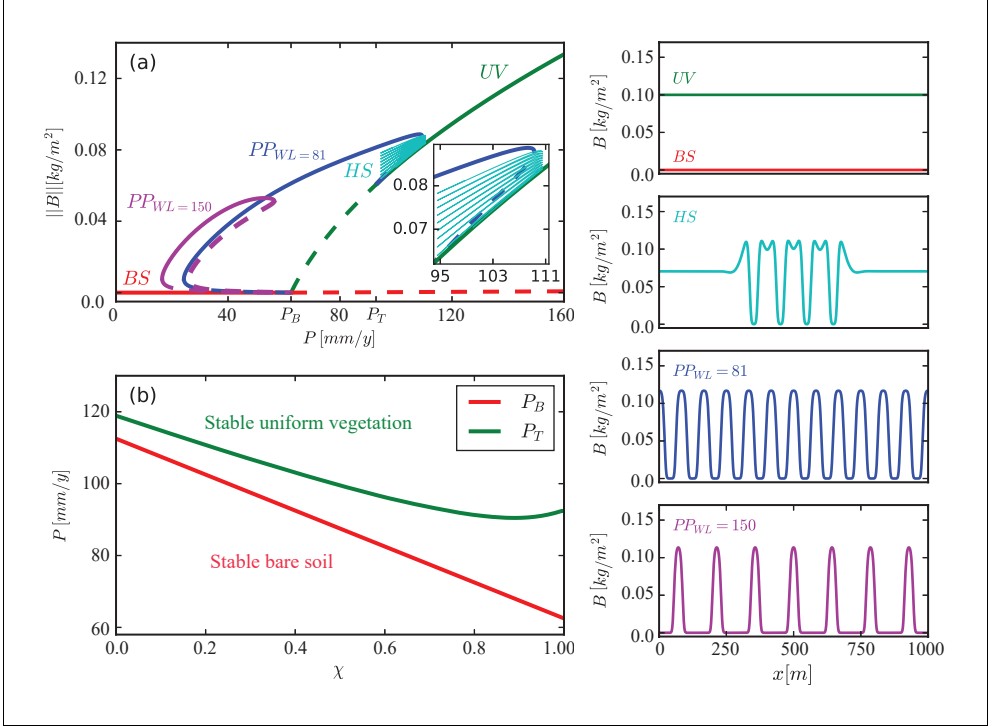

**Figure 4.** Existence and stability ranges of various solutions of a model for a single functional group. (a) A bifurcation diagram showing the $L2$-norm of the biomass density vs. precipitation for $\chi = 1$. The colors and corresponding labels denote the different solution branches: uniform vegetation ($UV$), periodic patterns at different wavelengths ($PP$), hybrid states consisting of pattern domains in otherwise uniform vegetation ($HS$), and bare soil ($BS$). Solid (dashed) lines represent stable (unstable) solutions. Example of spatial profiles of these solutions are shown in the insets on the right. (b) Instability thresholds of uniform vegetation, $P_T$, and of bare soil, $P_B$, as functions of $\chi$.

also imply that species investing in growth are more at risk of collapse to bare soil as a result of disturbances. This is because of the wider bistability range of vegetation patterns and bare soil, which makes these species more vulnerable to disturbances that shift the system to the alternative stable bare-soil state, such as overgrazing.

## Effects of spatial patterning on community assembly

What forms of community assemblages can emerge when the $N$ functional groups are allowed to interact and compete with one another? The asymptotic biomass distribution shown in *Figure 3b* has been calculated for a spatially uniform community, where all functional groups asymptotically form uniform vegetation ($P > P_T(\chi \to 0)$, see *Figure 4b*). To study the response of plant communities to progressively drier climates, we calculated the asymptotic biomass distributions for decreasing precipitation values. As *Figure 5* shows, at $P_1 = 150$ mm/y (panel a), a spatially uniform community develops, characterized by a symmetric hump-shape distribution of functional groups around a most abundant group at $\chi_{max} = \chi_0 = 0.62$, and by functional richness $FR_0 = 0.29$. Lowering the precipitation to $P_2 = 100$ mm/y (panel b), results in a spatially uniform community shifted to species that better tolerate water stress (higher $\chi$), now distributed around a most abundant functional group at $\chi_0 = 0.78$, and reduced functional richness, $FR_0 = 0.25$.

Lowering the precipitation further yet to a value, $P_3 = 80$ mm/y, below the Turing threshold $P_T$, results in spatial patterning (panel c). Interestingly, the community composition is now shifted back to species that favor investment in growth (lower $\chi$), and the functional richness increases rather than continue to decrease; the patterned community is distributed around $\chi_{k_1} = 0.68$ and its functional richness is $FR_{k_1} = 0.27$. Panel (c) also shows (in black) the biomass distribution of the unstable spatially-uniform community at $P_3 = 80$ mm/y, which continues the trend of panels (a,b) and demonstrates the significant change in community structure that spatial self-organization induces.

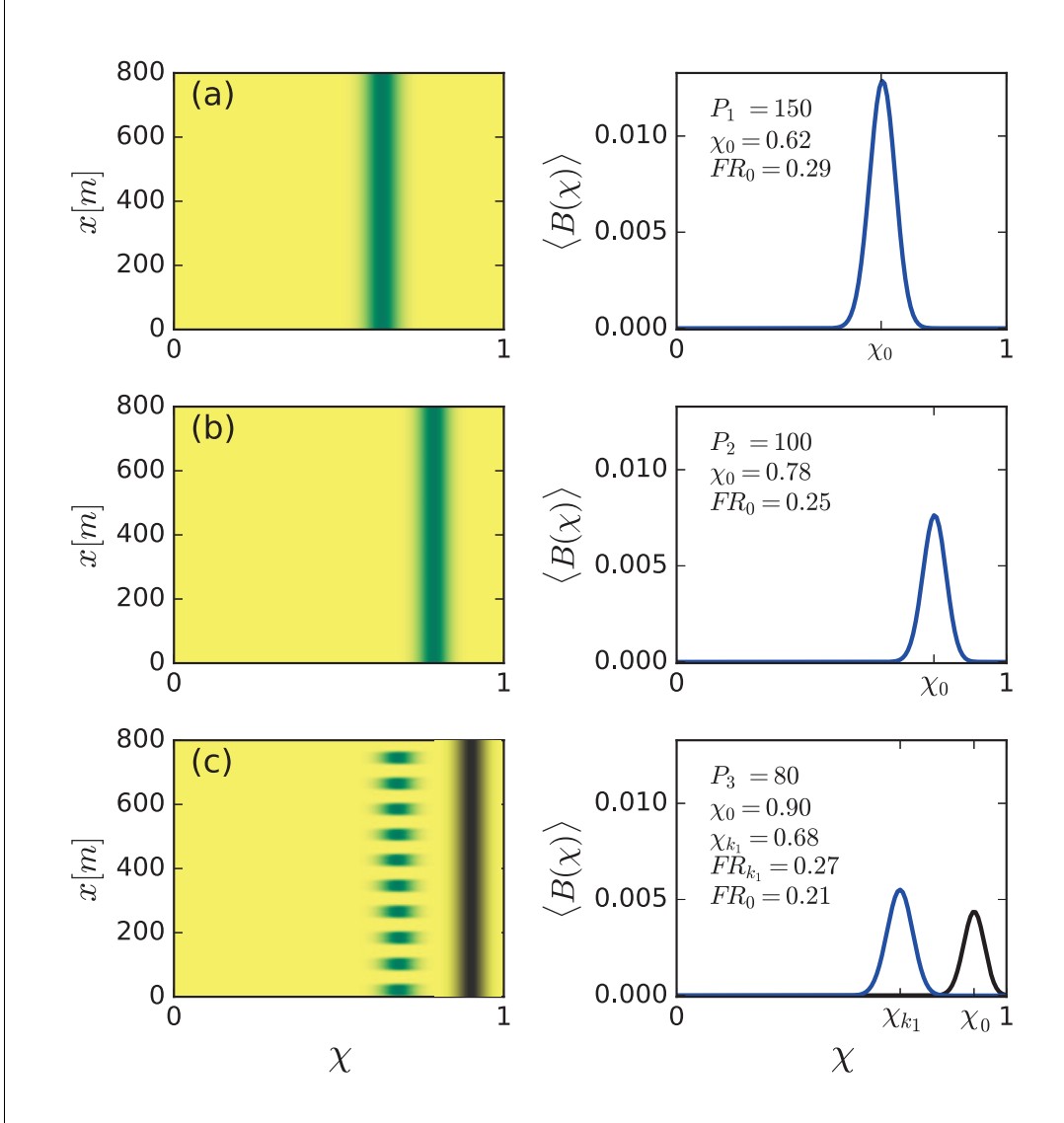

**Figure 5.** Community reassembly in response to precipitation downshifts. Left panels show biomass distributions in the trait ($\chi$) – space ($x$) plane for the specified precipitation rates $P_1, P_2, P_3$. Right panels show biomass profiles along the $\chi$ axis averaged over space. (a,b) A precipitation downshift from $P = 150$ mm/y to 100 mm/y, starting with a uniform community, results in a uniform community shifted to more tolerant species (higher $\chi$), and of lower functional richness (*FR*). (c) Further decrease to $P = 80$ mm/y results in a patterned community that is shifted back to species investing in growth (lower $\chi$), and has higher functional richness. The biomass distributions in black refer to the unstable uniform community.

Furthermore, self-organization in spatial patterns increases the total biomass as can be seen by comparing the areas enclosed by the biomass-distribution curves for uniform (black curve) and patterned (blue curve) communities in *Figure 5c*.

Once periodic patterns form, a further decrease in precipitation does not result in significant community-structure changes, unlike the case of uniform communities, as *Figure 6* shows. While spatially uniform communities move to higher $\chi$ values with decreasing precipitation, as the monotonically decreasing graph $\chi_{max} = \chi_0(P)$ shows, spatially patterned communities remain largely unchanged, as the nearly horizontal graphs $\chi_{max} = \chi_{k_1}(P)$ and $\chi_{max} = \chi_{k_2}(P)$ show. The first graph represents the periodic pattern that emerges at the Turing instability point $P_T$. Along this graph, as $P$ is reduced, the number of vegetation patches remains constant, but their size (along the $x$ axis) significantly reduces (compare the insets at $P = P_3$ and $P = P_4$ in *Figure 6*). Furthermore, the patches span the same range of functional groups (patch extension along the $\chi$ axis), that is, retain their functional

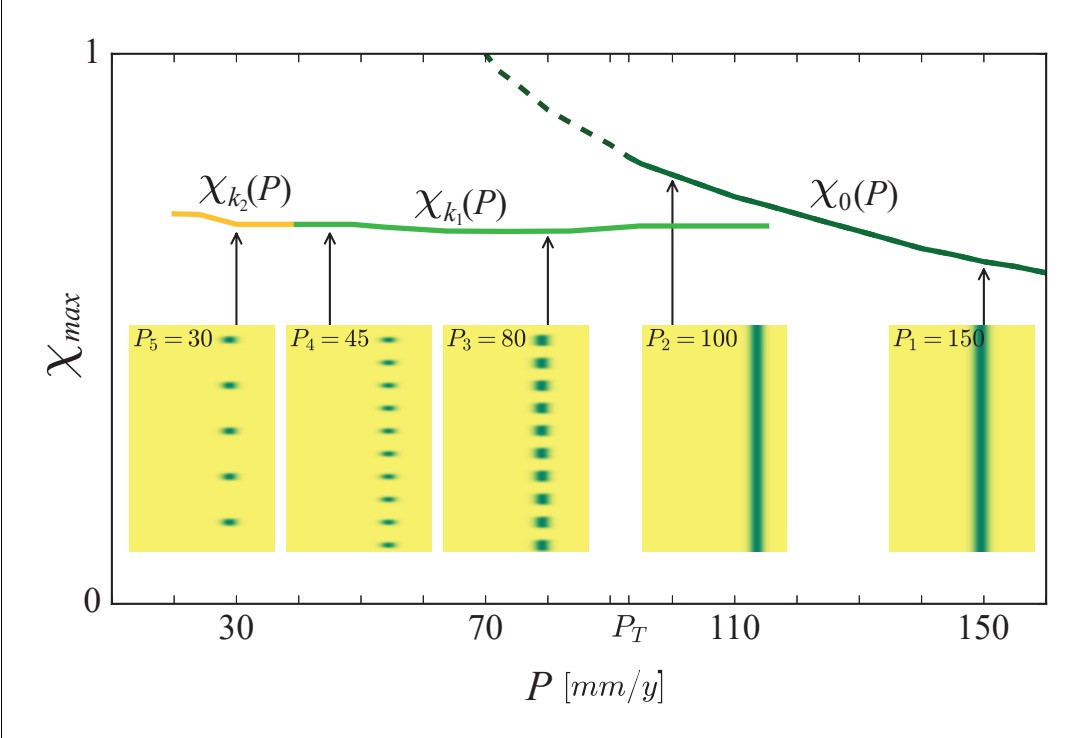

**Figure 6.** The buffering effect of spatial patterning on community structure. Shown is a partial bifurcation diagram depicting different forms of community assembly along the precipitation axis, as computed by integrating the model equations in time. Stable, spatially uniform communities, $\chi_0(P)$ (solid dark-green line), shift to stress-tolerant species (higher $\chi$), as precipitation decreases. When the Turing threshold, $P_T$, is traversed, spatial self-organization shifts the community back to fast-growing species (lower $\chi$), and keep it almost unaffected as the fairly horizontal solution branches, $\chi_{k_1}(P)$ (light green line), representing periodic patterns of wavenumber $k_1$, and $\chi_{k_2}(P)$ (yellow line), representing patterns of lower wavenumber $k_2$, indicate. The insets show biomass distributions in the ($\chi$, $x$) plane for representative precipitation values. The unstable solution branch describing uniform vegetation (dashed line) was calculated by time integration of the spatially decoupled model.

richness, and their most abundant functional group, $\chi_{max} = \chi_{k_1}(P)$, does not change. Thus, as $P$ is reduced, the patterned community that emerges at the Turing instability hardly changes in terms of pattern wavenumber $k_1$ or wavelength $2\pi/k_1$, and in terms of community structure, but the abundance of all functional groups reduces significantly as the vegetation patches become thinner along the $x$ axis (*Yizhaq et al., 2005*; *Siteur et al., 2014*).

The second graph in *Figure 6*, $\chi_{max} = \chi_{k_2}(P)$, represents a periodic vegetation pattern consisting of fewer patches (along the $x$ axis). Their extension along the $\chi$ axis remains approximately constant (functional richness hardly changes), and the same holds for the most abundant functional group, $\chi_{max} = \chi_{k_2}(P)$. Thus, the effect of further precipitation decrease is a transition to periodic pattern of longer wavelength and reduced productivity (*Siteur et al., 2014*; *Bastiaansen and Doelman, 2019*), but the community structure (composition and richness) remains almost unaffected.

## Effects of multistability on community assembly

The partial bifurcation diagram of *Figure 4*, obtained for a single functional group, indicates the existence of precipitation ranges where two or more stable states coexist. The significance of these multistability ranges lies in the possible existence of spatial mixtures of different states as additional stable states, and in the possibility of shifting a given stable state to an alternative stable state that is more desired. An example of the former case is the range where both uniform vegetation and periodic patterns are stable. In a sub-range of this range – the so-called snaking range (*Knobloch, 2015*) – a multitude of additional non-periodic hybrid states exist (denote by HS in *Figure 4a*; *Meron, 2019*). An example of the latter case is a range where two stable periodic patterns of widely different wavelengths coexist. *Figure 4* shows only two solution branches of this kind, but many more exist within a range of wavelengths for a given precipitation value that define the so-called

Busse balloon (*Sherratt, 2013*; *Zelnik et al., 2013*; *Siteur et al., 2014*; *Bastiaansen et al., 2018*; *Bastiaansen et al., 2020*). We focus here on the former case and, specifically, on hybrid states.

The effect of hybrid states on community structure is demonstrated in *Figure 7*. Shown are three examples of hybrid community states that differ in the size of the patterned domain relative to the uniform domain. These states were obtained as asymptotic solutions of the community model *Equations (1)*, starting from similar hybrid solutions of a single functional-group model as initial conditions (*Figure 4*). The community dynamics that develop in the course of time lead to niche differentiation; stress-tolerant species residing in the uniform domains, and fast growing species residing in the patterned domains. As a consequence, the global, whole-system functional richness increases significantly, compared to the richness associated with a purely uniform state or a purely patterned state, as the right panels in *Figure 7* indicate (compare with *Figure 5b*). Furthermore, the relative size of the patterned domains affects the functional evenness, *FE* (see *Equation 6*). A relatively high *FE* value is obtained for uniform and patterned domains of comparable sizes (*Figure 7b*) and low evenness when the relative size of the patterned domain is either small or large (*Figure 7a,c*).

## Discussion and conclusion

The results described above provide three insights into the intimate relationships between spatial self-organization, community assembly and ecosystem management, as illustrated in *Figure 1* and explained below.

### Insight I: Spatial patterning acts to reverse community-structure changes induced by environmental stress

According to *Figure 5*, reduced precipitation shifts spatially uniform communities to stress-tolerant species (higher $\chi$ values), but when the Turing threshold is traversed and self-organization in periodic spatial patterns occurs, this trend is reversed and a shift back to species investing more in growth and less in stress tolerance takes place. This surprising change of community structure reflects the complex nature of ecosystem response to varying environments, which can employ mechanisms operating in parallel at different organization levels. The composition shift to higher $\chi$ values, as $P$ decreases but still remains above the Turing threshold $P_T$, is driven by community-level processes, whereby *inter*specific competition results in a community consisting of species that are better adapted to water stress, and of lower functional richness. By contrast, the composition shift to lower $\chi$ values, once the Turing threshold is traversed, is driven by population-level processes of spatial self-organization, whereby *intra*specific competition results in partial mortality and the appearance of bare-soil patches. These patches provide an additional source of water to adjacent vegetation patches, besides direct rainfall, through overland water flow (*Meron, 2019*). That additional resource compensates for the reduced precipitation and relaxes the local water stress at vegetation patches. The resulting ameliorated growth conditions favor species investing in growth (lower $\chi$), and increase functional richness.

### Insight II: Spatial re-patterning buffers community-structure changes

Once periodic patterns form, a further decrease of precipitation does not result in significant community-structure changes, as *Figure 6* shows. This is because of additional forms of spatial self-organization that buffer the impact of decreasing precipitation. The first of which is partial plant mortality that results in vegetation patches of reduced size (compare the insets at $P = P_3$ and $P = P_4$ in *Figure 6*). These patches benefit from increased water availability due to the larger water-contributing bare-soil patches that surround them. This response form does not involve a change in the number of patches or pattern's wavenumber, and occurs along the branch of any periodic solution, including those shown in *Figure 6*, $\chi_{k_1}(P)$ and $\chi_{k_2}(P)$. A second form of spatial self-organization involves plant mortality that results in patch elimination and wavenumber reduction (*Yizhaq et al., 2005*; *Siteur et al., 2014*), such as the transition from $\chi_{k_1}(P)$ to $\chi_{k_2}(P)$ in *Figure 6* (see insets at $P = P_4$ and $P = P_5$). Like in the first form, any remaining vegetation patch benefits from larger bare-soil patches surrounding it and, thus, from higher water availability.

These forms of spatial self-organization apply also to two-dimensional systems, especially to gently sloped terrains, where quasi-one-dimensional patterns of stripes oriented perpendicular to

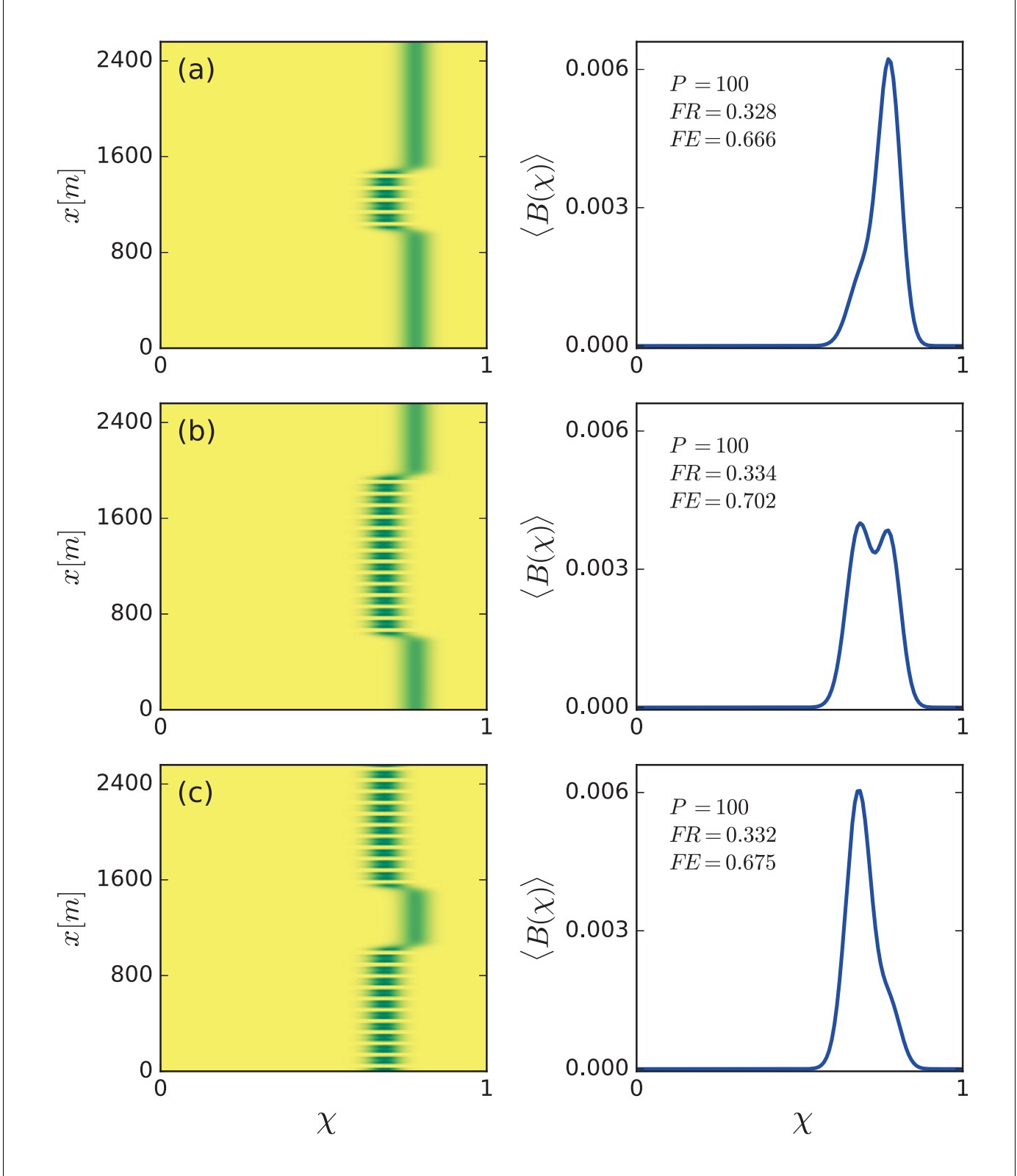

**Figure 7.** Increased functional diversity of hybrid states and evenness control. Left panels show biomass distributions of different hybrid states in the trait ($\chi$) – space ($x$) plane. Right panels show biomass profiles along the $\chi$ axis averaged over space. The functional richness, *FR*, of all hybrid states is almost equal and higher than that of purely uniform or purely patterned states (compare with panel b in **Figure 5**), but their functional evenness, *FE*,

*Figure 7 continued on next page*

*Figure 7 continued*

differs – high for patterned and uniform domains of comparable sizes (**b**) and low for small (**a**) and large (**c**) pattern-domain sizes. Calculated for a precipitation rate $P = P_2 = 100$ mm/y.

the slope direction occupy wide precipitation ranges (*Deblauwe et al., 2010*) and are widely observed in nature (*Valentin et al., 1999*; *Deblauwe et al., 2012*; *Bastiaansen et al., 2018*). Two-dimensional systems, however, allow for additional forms of patterning and re-patterning, which we have not studied in this work (*von Hardenberg et al., 2001*; *Rietkerk et al., 2002*; *Lejeune et al., 2004*; *Gowda et al., 2014*; *Meron, 2019*). In flat terrains uniform vegetation responds to reduced precipitation, below the Turing threshold $P_T$, by forming hexagonal gap patterns. These patterns consist of periodic arrays of circular bare-soil gaps, where any gap is surrounded by six other equidistant gaps. Further decrease in precipitation, below a second threshold, results in a morphology change, where bare soil gaps grow and merge to form patterns of parallel stripes or labyrinthine patterns. Below a third precipitation threshold, a second morphology change takes place, where vegetation stripes breakup to form hexagonal spot patterns. These patterns consist of periodic arrays of circular vegetation spots, where any spot is surrounded by six other equidistant spots.

The common denominator underlying these morphology changes is the increase in bare-soil areas adjacent to vegetation patches, as precipitation is reduced; from bare-soil gaps to bare-soil stripes in the first morphology change, and from bare-soil stripes to bare-soil areas surrounding vegetation spots in the second morphology change. Increasing bare-soil areas compensate for the reduced precipitation by providing an extra source of water – water transport to adjacent vegetation patches through overland water flow, soil-water diffusion, or water conduction by laterally extended roots. These processes act to retain the amount of water available to vegetation patches and buffer the impact of decreased precipitation. As a consequence, and like the one-dimensional case discussed earlier, community structure is expected to remain largely unaffected.

## Insight III: Multiplicity of stable community states and ecosystem management

The multitude of stable hybrid states, that is patterned domains of different sizes in otherwise uniform vegetation, open up novel directions for sustainable management of stressed ecosystems that integrate the need for provisioning ecosystem services with the need to preserve species diversity, as explained below.

A spatially uniform community state at high precipitation responds to a drier climate by shifting the community composition to stress-tolerant (high $\chi$) species and reducing its functional richness (see *Figure 5a,b*). Ecosystem services, such as feeding livestock by grazing, impose further stress and are likely to result in further reduction of functional richness. However, a sufficiently drier climate also induces a Turing instability to periodic patterns and a multitude of stable hybrid states. These states have higher functional richness than those of the uniform and patterned states, separately, and can be even higher ($FR \approx 0.33$) than the functional richness of the original uniform community state before the shift to a drier climate has occurred ($FR = 0.29$, see *Figure 5a*). This is because uniform domains give rise to stronger competition and higher water stress, and thus form niches for stress-tolerant species, whereas patterned domains benefit from overland water flow from bare-soil patches, which weakens the competition, and provides niches to fast-growing species.

These results can be used to reconcile the conflicting needs for ecosystem services and preservation of functional diversity, through the management of provisioning ecosystems services by non-uniform biomass removal, so as to induce the formation of patterned domains or hybrid states (*Figure 1c*). Moreover, the multitude of stable hybrid states allow to control the relative abundance of fast growing vs. stress tolerant species, and thus the functional evenness, $FE$, of the community; small patterned domains in uniform vegetation (*Figure 7a*) or small uniform domains in patterned vegetation (*Figure 7c*) give rise to low functional evenness ($FE \approx 0.67$), while domains of comparable sizes give rise to higher evenness ($FE \approx 0.70$).

Hybrid states exist within the bistability precipitation range of uniform and patterned vegetation in a subrange where front solutions separating domains of these alternative stable states are stationary (pinned) (*Pomeau, 1986*; *Knobloch, 2015*; *Meron, 2019*). Thus, observations of such fronts that

remain stationary in the course of time provide indications for the possible realizations of hybrid states. Although the existence range of hybrid states is not large, temporal rainfall varibility that occasionally takes the system outside this range may have little effect on community structure as front speeds are expected to be low.

The family of periodic patterns of different wavelengths that exist along the rainfall gradient and their wide ranges of multistability (of which only two are shown in *Figure 4*) *Zelnik et al., 2013*; *Siteur et al., 2014*, suggest another sustainable form of managing provisioning ecosystem services involving biomass removal – pattern dilution by patch removal, for example every second patch. Shifting in this manner a periodic pattern of a given wavelength to an alternative pattern of longer wavelength can result in significantly larger patches and, consequently, in a gradient of water stress; high stress in the patch center and low stress in the patch edge. This is because most of the overland water that flows toward the patch infiltrates at the patch edge. This stress gradient, in turn, should result in niche differentiation and a more diverse community with stress tolerant species residing in the patch center and fast-growing species in the patch edges. However, further studies of periodic solutions of the community model are needed to substantiate this suggestion.

### Concluding remarks

In this work, we studied the interplay between spatial self-organization and community reassembly, using a spatial model of dryland plant communities. This model captures a particular pattern-forming feedback, associated with differential infiltration and overland water flow (*Figure 2*), but we expect our findings to hold also for models that capture the feedbacks associated with soil-water diffusion, and water conduction by laterally spread roots, since they all lead to the same bifurcation structure (*Figure 4a*). We also considered a particular tradeoff, but expect our results to hold for other trade-offs as well, such as investment in aboveground vs. belowground biomass (*Nathan et al., 2016*). We are not aware of empirical studies of dryland ecosystems that address this interplay, and thus of data that can be used to test our theoretical predictions, but the three insights described above can serve as solid hypotheses for new long-term empirical studies.

Spatial self-organization is not limited to dryland ecosystems. Periodic and non-periodic vegetation patterns have been observed and studied in a variety of other ecological contexts including non-drylands plant communities with negative plant-soil feedbacks, such as self-toxicity (*Bonanomi et al., 2014*; *Marasco et al., 2014*), hydric peat bogs (*Weltzin et al., 2000*; *Eppinga et al., 2008*), seagrass meadows (*Ruiz-Reynés et al., 2017*), salt marshes (*Zhao et al., 2019*; *Zhao et al., 2021*), aquatic macrophytes in stream ecosystems (*Cornacchia et al., 2018*), and others (*Rietkerk and van de Koppel, 2008*). In most of these systems spatial self-organization inter-mingles with community dynamics; the findings of this work may be relevant to these contexts as well, or motivate new studies along similar lines.

## Materials and methods

We use a continuum modeling approach where plant populations are described by above-ground biomass densities rather than by plant-number densities. This approach allows a continuum representation even of highly discrete plant populations, as often encountered in arid regions (*Meron et al., 2019*). The model is based on a model platform that has been introduced and discussed in earlier studies (*Gilad et al., 2004*; *Gilad et al., 2007*; *Nathan et al., 2016*; *Meron, 2016*), but involves several modifications. It consists of a system of partial differential equations (PDEs) for biomass variables, $B_i$ ($i = 1, ..., N$), representing the spatial densities of above-ground biomass of the $N$ functional groups, $\chi_1, \chi_2, ..., \chi_N$, and two water variables, $W$ and $H$, representing spatial densities of below-ground and above-ground water, respectively, all in units of kg/m$^2$. We restrict ourselves in this work to one-dimensional spatial dependence of all state variables. Thus, solutions that are periodic in the $x$ direction are assumed to be independent of the orthogonal $y$ direction and represent periodic stripe patterns. The community model then reads:

$$\partial_t B_i = \Lambda_i W B_i - M_i B_i + D_B \partial_x^2 B_i + D_\chi \partial_\chi^2 B_i \tag{1a}$$

$$\partial_t W = IH - LW - \Gamma W \sum_{j=1}^{N} B_j + D_W \partial_x^2 W \tag{1b}$$

$$\partial_t H = P - IH + D_H \partial_x (H^\alpha \partial_x H) \tag{1c}$$

where the second 'trait derivative', $\partial_\chi^2 B_i \equiv N^2(B_{i+1} - 2B_i + B_{i-1})$, represents mutations at a very small rate $D_\chi$. In these equations, the growth rate of the $i$ th functional group, $\Lambda_i$, the infiltration rate of above-ground water into the soil, $I$, and the evaporation rate, $L$, are given by the expressions:

$$\Lambda_i = \frac{\Lambda_0 K_i}{\bar{B} + K_i}, \quad I = \frac{A(\bar{\bar{B}} + fQ)}{\bar{\bar{B}} + Q}, \quad L = \frac{L_0}{1 + R\bar{B}}, \tag{2}$$

where $\bar{B} = \sum_{j=1}^{N} B_j$ and $\bar{\bar{B}} = \sum_{j=1}^{N} Y_j B_j$. We note that the biomass variable $B_i$ represents the above-ground biomass of all species with functional traits $K_i, M_i$ within the interval $\Delta\chi$ that precedes $\chi_i$. In other words, while $B_i$ represents a biomass density in physical space, it does not represent a density in trait space, that is biomass per unit trait-space length. In this approach the solutions of the model *Equations (1)* depend on the choice of $N$. Specifically, the amplitude of the biomass distribution decreases as $N$ increases, although the metrics $\chi_{max}, FR, FE$ are practically independent of $N$ for sufficiently large $N$ values. An alternative approach is to express the model equations in terms of the biomass densities $b_i = B_i/\Delta\chi = NB_i$. Expressed in terms of $b_i$, also the biomass distributions are independent of $N$. That alternative model presentation, where $B_i$ is replaced by $b_i\Delta\chi$, is also convenient when considering the continuum limit $N \to \infty$; then $b_i\Delta\chi$ becomes $b_i d\chi$ and the summation over the $N$ functional groups should simply be replaced by an integral.

A list of all model parameters, their descriptions, units and their values are given in *Table 1*. Below, we focus on the new elements in the model and on several aspects we wish to emphasize. The biomass dependence of the growth rates, $\Lambda_i$, models competition for light and accounts for growth attenuation due to shading. That attenuation is described by the parameters $K_i$, which quantify the capacity of functional groups to capture light; high $K_i$ values represent plant species investing preferably in shoot growth that are less affected by shading. Note that the growth rate also includes attenuation due to self-shading (*Noy-Meir, 1975*). This attenuation form represents a relative decrease in the photosynthetic capacity of a plant, as photosynthesis is progressively limited to the upper layer of leaves. The parameter $\Lambda_0$ represents the growth rate at low biomass values for which competition for light is absent.

The biomass dependence of the infiltration rate, $I$, is responsible for differential infiltration, quantified by the dimensionless parameter $0 \leq f \leq 1$ and the parameter $Q$. Low $f$ values represent highly differential infiltration, low in bare soil ($I = fA$) and high ($Q$-dependent) in vegetation patches, and constitute an important element in the pattern-forming feedback associated with overland water flow toward denser vegetation patches (*Meron, 2019*). We assume that all species contribute to this differential infiltration, but these contributions are not necessarily equal as they depend on various factors, including the number density of individual plants and their root densities. We quantify these differences through the parameters $Y_i$.

The biomass dependence of the evaporation rate, $L$, accounts for reduced evaporation in vegetation patches due to shading, quantified by the parameter $R$. The parameter $L_0$ represents evaporate rate in bare soil. The reduced evaporation in vegetation patches is a positive feedback between biomass and water that can result in bistability of bare soil and uniform or patterned vegetation. This feedback is not scale dependent, as it does not involve water transport, and therefore cannot induce spatial patterning.

Tolerance to water stress is modeled through the mortality parameters $M_i$ – a plant investment in tolerating stress reduces the mortality rate. This parameter may lump together (as additional additive terms) other biomass-decay factors, such as grazing stress. For simplicity we choose $M_i$ to be independent of $W$ (unlike in *Tzuk et al., 2019* for example). This choice can represent stress-tolerance mechanisms associated with plant architecture rather than phenotypic changes, such as hydraulically independent multiple stems that lead to a redundancy of independent conduits and higher resistance to drought (*Schenk et al., 2008*). We note that although $M_i$ is independent of $W$, lower

**Table 1.** Model paremeters, their descriptions, numerical values and units.

| Parameter | Description | Value | Unit |
|---|---|---|---|
| $\Lambda_0$ | Growth rate at zero biomass | 0.032 | $m^2/(kg \cdot y)$ |
| $\Gamma$ | Water uptake rate | 20.0 | $m^2/(kg \cdot y)$ |
| $f$ | Infiltration contrast ($f \ll 1$ – high contrast) | 0.01 | - |
| $A$ | Maximal value of infiltration rate $I$ | 40.0 | $y^{-1}$ |
| $Q$ | Reference biomass at which $I \approx A/2$ for $f \ll 1$ | 0.06 | $kg/m^2$ |
| $L_0$ | Evaporation rate in bare soil | 4.0 | $y^{-1}$ |
| $R$ | Evaporation reduction due to shading | 10.0 | $m^2/kg$ |
| $K_i$ | Capacity to capture light | variable | $kg/m^2$ |
| $K_{min}$ | Minimal capacity to capture light | 0.1 | $kg/m^2$ |
| $K_{max}$ | Maximal capacity to capture light | 0.6 | $kg/m^2$ |
| $M_i$ | Mortality rate | variable | $y^{-1}$ |
| $M_{min}$ | Minimal mortality rate | 0.5 | $y^{-1}$ |
| $M_{max}$ | Maximal mortality rate | 0.9 | $y^{-1}$ |
| $Y_i$ | Relative contribution to infiltration rate | variable | - |
| $Y_{min}$ | Minimal contribution to infiltration rate | 0.5 | - |
| $Y_{max}$ | Maximal contribution to infiltration rate | 1.5 | - |
| $P$ | Precipitation rate | variable | $mm/y$ |
| $\chi$ | Tradeoff parameter | [0,1] | - |
| $N$ | Number of functional groups | 128 | - |
| $D_B$ | Biomass dispersal rate | 1.0 | $m^2/y$ |
| $D_W$ | Soil-water diffusion coefficient | $10^2$ | $m^2/y$ |
| $D_H$ | Overland-water diffusion coefficient | $10^4$ | $m^2/y$ |
| $D_\chi$ | Trait diffusion rate | $10^{-6}$ | $y^{-1}$ |

values of $M_i$ do represent higher tolerance to water stress. This can be seen by considering the fitness of the $i$ th functional group $\Lambda_i W - M_i$, which implies that functional groups with lower mortality tolerate lower soil-water content or higher water stress, as their fitness can remain positive.

Note that the atmosphere is considered in this model as the ecosystem's environment, rather than a system part in feedback relationships with vegetation growth. As a consequence, it is quantified by a precipitation parameter $P$ in the equation for $H$ (1 c), assumed in this study to be constant. This simplification is justifiable in drylands where vegetation is relatively sparse and the overall transpiration is low. Also, for simplicity, we choose to describe here overland water flow as a linear diffusion problem by setting $\alpha = 0$ in *Equations (1)*. Although that process is nonlinear, the qualitative results and conclusions reported here do not depend on that choice (see *Gilad et al., 2004* for the choice $\alpha = 1$).

As pointed out earlier, we distinguish between different species through the different tradeoffs they make between shoot growth and tolerance to water stress. We capture this tradeoff using the parameters $K_i$ and $M_i$ through the tradeoff relations:

$$K(\chi) = K_{max} + \chi(K_{min} - K_{max})$$
$$M(\chi) = M_{max} + \chi(M_{min} - M_{max}), \tag{3}$$

where $K_i = K(\chi_i)$ and $M_i = M(\chi_i)$. According to these relations, $\chi \to 0$ represents the functional group $(K_{max}, M_{max})$ with highest investment in growth and lowest investment in tolerance (highest mortality), while $\chi = 1$ represents the functional group $(K_{min}, M_{min})$ with lowest investment in growth and highest investment in tolerance. This tradeoff is likely to affect the contributions, $Y_i$, of the various functional groups to the infiltration rate $I$; denser roots associated with lower-$\chi$ species (increased investment

in shoot growth) make the soil more porous and increase the infiltration rate. An additional contribution to that effect is lower soil-crust coverage in patches of lower-$\chi$ species. We therefore assume the following form for $Y_i$:

$$Y_i = Y(\chi_i) = Y_{max} + \chi_i(Y_{min} - Y_{max}). \tag{4}$$

The model for the single functional group $\chi = 1$ used in deriving the results of *Figure 4* is obtained from the community model (1) by setting all biomass variables identically to zero, apart of $B_N$ for which $\chi = \chi_N = N\Delta\chi = 1$.

We measure functional diversity using two metrics, functional richness, *FR*, and functional evenness, *FE* (*Mason et al., 2005*). The first metric is given by the extent of the biomass distribution around $\chi_{max}$, as *Figure 3* illustrates. The second metric contains information about the abundance of functional groups in the community and how even the abundance is among the groups. We use here the following analogue of the Shannon diversity index,

$$H' = -\sum_{i=1}^{N} b_i \ln b_i, \qquad b_i = \frac{B_i}{\sum_{j=1}^{N} B_j}, \tag{5}$$

and the related index of Pielou for functional evenness *Pielou, 1966*,

$$FE = H'/\ln N. \tag{6}$$

We use numerical continuation methods (AUTO *Doedel, 1981*) to study the model equations for single functional groups, and numerical time-integration in the composed trait-space plane to study the full community model. We use periodic boundary conditions in $x$ and zero-flux conditions in $\chi$. Initial conditions are chosen to contain all functional groups, even if at arbitrarily small biomass values. Such small values represent the presence of seeds that remain viable even when they cannot germinate (*DeMalach et al., 2021*). Source-code files for integrating the community and single functional-group models are available at the GitHub repository (*Bera, 2021*).

We wish to point out that the results presented here are not sensitive to the particular choice of the parameter values displayed in *Table 1*, and similar results have been obtained with other sets of parameter values. The main constraint on this choice is the need to capture the Turing instability of the uniform vegetation state, and the need to define the tradeoff relations such that the Turing threshold split the community into functional groups that form and do not form periodic patterns. Since vegetation patterns are observed on spatial scales that differ by orders of magnitude, from periodicity of tens of centimeters for herbaceous vegetation to periodicity of tens of meters for woody vegetation (*Rietkerk et al., 2004*), we focus on generic community aspects associated with vegetation patterning, rather than attempt to model a particular ecosystem with a specific spatial scale.

## Acknowledgements

We thank the reviewers Gianalberto Losapio and Mara Baudena, and the anonymous reviewer, for their constructive criticisms and helpful comments. The research leading to these results has received funding from the Israel Science Foundation under Grant No. 1053/17. During this research BKB has been supported by a PBC postdoctoral fellowship, and JJRB has been supported by a Kreitman postdoctoral fellowship.

## Additional information

### Funding

| Funder | Grant reference number | Author |
|---|---|---|
| Israel Science Foundation | 1053/17 | Ehud Meron |
| Planning and Budgeting Committee of the Council for Higher Education of Israel | Postdoctoral Fellowship | Bidesh K Bera |

Kreitman                    Postdoctoral Fellowship          Jamie JR Bennett

The funders had no role in study design, data collection and interpretation, or the decision to submit the work for publication.

## Author contributions
Bidesh K Bera, Formal analysis, Investigation, Methodology, Project administration, Writing - review and editing; Omer Tzuk, Conceptualization, Formal analysis, Investigation, Methodology, Writing - review and editing; Jamie JR Bennett, Formal analysis, Investigation, Methodology, Writing - review and editing; Ehud Meron, Conceptualization, Supervision, Funding acquisition, Investigation, Methodology, Writing - original draft, Writing - review and editing

## Author ORCIDs
Omer Tzuk (iD) http://orcid.org/0000-0002-6541-3311
Jamie JR Bennett (iD) https://orcid.org/0000-0002-9748-5010
Ehud Meron (iD) https://orcid.org/0000-0002-3602-7411

## Decision letter and Author response
Decision letter https://doi.org/10.7554/eLife.73819.sa1
Author response https://doi.org/10.7554/eLife.73819.sa2

# Additional files

## Data availability
The current manuscript is a computational study, so no empirical data have been generated for this manuscript. Modelling code is uploaded to https://github.com/bidesh001/Plant-community-model (copy archived at https://archive.softwareheritage.org/swh:1:rev:976242ce55d8f501d25cb67c365004f04b895b6c).

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
