## [Decision Letter]

**Acceptance summary:**

In this paper, the authors use a trait-based model of plant growth and water flow in drylands to show that under increasing water shortage, spatial self-organization can help plant communities to maintain biodiversity and thus ecosystem functioning. Spatially heterogeneous ecosystem management may support these processes.

**Decision letter after peer review:**

[Editors’ note: the authors submitted for reconsideration following the decision after peer review. What follows is the decision letter after the first round of review.]

Thank you for submitting the paper "Linking spatial self-organization to community assembly and biodiversity" for consideration by *eLife*. Your article has been reviewed by 3 peer reviewers, and the evaluation has been overseen by a Reviewing Editor and a Senior Editor. The following individuals involved in review of your submission have agreed to reveal their identity: Gianalberto Losapio (Reviewer #2); Mara Baudena (Reviewer #3).

While we cannot go forward with the current version of the manuscript, since revision would almost certainly take substantially longer than allowed per *eLife* policy, we remain very interested in the work. We would therefore welcome a substantially revised version, which we would treat as a new submission, but aim to recruit the same editors and reviewers.

*Reviewer #1:*

Bera et al. study the response of vegetation in water-limited ecosystems to changes in the precipitation regime. Previous studies have shown that spatial processes, in particular the redistribution of (soil and surface) water, may play an important role in mediating the ecosystem response. An important consequence of this redistribution is the spatial self-organization of vegetation into regular spatial patterns, consisting of vegetation patches that act as sinks for (surface) water, and surrounding areas of bare soil that act as water sources. At the ecosystem level, the additional water input in vegetation patches may enable vegetation to persist at precipitation levels that are too low to sustain a spatially uniform cover.

While most model studies of spatial self-organization and pattern formation describe vegetation dynamics through 1-2 biomass variables, the current study extends this previous work by considering a trait diversity gradient, considering a large number (N=128) of discrete trait classes that range from stress-tolerant to fast-growing characteristics. The results show that in the absence of spatial pattern formation, a decrease in precipitation leads to a shift in the biomass distribution toward the more stress-tolerant trait classes. At the onset of pattern formation, however, soil water availability increases at the locations where vegetation patches form, enabling the more fast-growing trait classes to increase in biomass, and this shift is accompanied by an increase in functional diversity of trait classes as well. It is also shown that once these patterned ecosystem states are formed, the main adaptation to further decreases in precipitation occurs either through shrinking the size of existing patches, or by reducing the number of patches; in contrast, biomass and community composition of the patches remains relatively stable. Finally, it is shown that for certain precipitation conditions, functional diversity is maximized when the ecosystem is in a hybrid state, where part of the landscape has a spatially uniform vegetation cover, and part of the landscape is in a patterned state.

A potential strength of this paper is that the community assembly and biodiversity perspective on spatial self-organization may highlight the relevance of pattern formation in ecosystems more clearly to a broad audience. The formulation of a trait/strategy gradient of discrete classes is certainly an interesting suggestion to connect the typical single/few biomass variable(s) approach to a community-level approach. The community assembly process is modelled in a very specific way, and the manuscript would benefit from an expanded ecological motivation of the processes that are being mimicked, and thereby explain more clearly what taxonomic level of organization is being considered. In addition, it would be useful if the authors could provide further clarification as to what extent the community diversity dynamics can be separated from total biomass dynamics of patterned water-limited ecosystems given the current approach. These points are explained in further detail below.

• First, it was not entirely clear to this reviewer how the reaction parts of the model equations determine the optimal trait value χ, and how this value varies as a function of precipitation. Assuming a single trait class, and plotting the relevant equilibrium values of the three state variables shed some light on this issue. [Unfortunately, there does not seem to be a possibility to attach the figure with these plots to this review report]. Assuming the non-spatial equilibrium solution was derived correctly , the optimum biomass (B) value shifts across the trait spectrum with changing precipitation (in the non-spatial model version, solving the surface water equation for equilibrium will always yield that all precipitation infiltrates, i.e. regardless of the values of surface water, H, and χ). The equilibrium of soil water availability (W), which is the growth limiting resource of the vegetation, shows an inverse pattern with biomass. This result is in line with a classical results (e.g. Tilman 1982), in that the most successful strategy is the one that is able to reduce the limiting resource to the lowest equilibrium value. With all trait classes competing for the soil water resource, however, it is then not immediately clear why the most successful trait class is not outcompeting the other classes. This leads to a second point, about the way in which community trait adaptation is modelled.

• The authors model trait adaptation through a diffusion approximation between trait classes. That is, every timestep, a small amount of biomass flows from the class with higher biomass to the neighboring trait class with lower biomass. From an ecological point of view, it seems that this process is describing adaptation of vegetation that is already present, so this process seems to be limited to intraspecific phenotypic plasticity. From the text, however, it seems that the trait classes correspond to higher taxonomic levels of organization, when describing shifts from fast growing to stress-tolerant species, for example. It is not entirely clear, however, how biomass flows as assumed in the model could occur at these higher levels of organization.

• Combining the observations from the previous two points, there is a concern that for a given level of precipitation, there is a single trait class with optimal biomass/lowest soil water level that is dominant, with the neighboring trait classes being sustained by the diffusion of biomass from the optimal class to neighboring inferior classes. This would seem a bit problematic, as it would mean that most classes are not a true fit for the environment, and only persist due to the continuous inflow of biomass. Taking a clue from the previous papers of the authors, it seems this may not be the case, though. Specifically, in the paper by Nathan et al. (2016) it seems that all trait classes are started at low initial biomass density, and the resulting steady state (in the absence of biomass flows between classes) seems to show similar biomass profiles as shown in Figures 4,5 and 7 of the current paper. While the current model formulation seems slightly different, similar results may apply here. Indeed, keeping all trait classes at non-zero (but low) density, and when the (abiotic and biotic) environment permits, let each class increase in biomass seems like the most straightforward approach to model community assembly dynamics. Given the above discussion about these trait classes competing for a single resource (soil water), and one trait class being able to drive this resource availability to the lowest level, it would then be useful to readers to explain why multiple trait classes can coexist here, and how (for spatial uniform solutions) the equilibrium soil water level with multiple trait classes present compares to the equilibrium soil water level when only the optimal trait class is present. Furthermore, if results as presented in Nathan et al. (2016) indeed hold in the current case, perhaps it means that the biomass profile responses as shown in e.g. Figure 5 would also occur if there was no biomass flow between trait classes included, but that the time needed to adjust the profile would take much longer as compared to when the drift term/second trait derivative is included. In summary, further clarification of what the biomass flows between classes represent, and the role it plays in driving the presented results would be useful for readers.

• In addition, it would be useful for readers to understand to what extent the shifts in average trait values and functional diversity can be decoupled from the biomass and soil water responses to changes in precipitation that would occur in a model with only a single biomass variable. For example, early studies on self-organization in semi-arid ecosystems already showed that the shift toward a patterned state involved the formation of patches with higher biomass, and higher soil water availability, as compared to the preceding spatially uniform state, and that the biomass in these patches remains relatively stable under decreasing rainfall, while their geometry changes (e.g. Rietkerk et al. 2002). It has also been observed that for a given environmental condition, biomass in vegetation patches tends to increase with pattern wavelength (e.g. Bastiaansen and Doelman 2018; Bastiaansen et al. 2018). Given the model formulation, one wonders whether higher biomass in the single variable model is not automatically corresponding to higher abundance of faster growing species and a higher functional diversity (as the diffusion of biomass can cover a broader range when starting from higher mass in the optimal trait class). There are some indications in the current work that the linkage is more complicated, for example, the biomass peak in Figure 7c is lower, but also broader as compared to the distribution of Figure 7b, but it is currently not entirely clear how this result can be explained (for example, it might be the case that in the spatially patterned states, the biomass profiles also vary in space).

• The possibility of hybrid states, where part of the landscape is in a spatially uniform state, while the other part of the landscape is in a patterned state, is quite interesting. To better understand how such states could be leveraged in management strategies, it would be useful if a bit more information could be provided on how these hybrid states emerge, and whether one can anticipate whether a perturbation will grow until a fully patterned state, or whether the expansion will halt at some point, yielding the hybrid state. It seems that being able to distinguish these case would be necessary in the design of planning and management strategies. Also, in Figure 3a, the region of parameter space in which hybrid states occur is not very large; it is not entirely clear whether the full range of hybrid states is left out here for visual considerations, or whether these states only occur within this narrow range in the vicinity of the Turing instability point.

References:

Bastiaansen R, Doelman A. 2018. The dynamics of disappearing pulses in a singularly perturbed reaction-diffusion system with parameters that vary in time and space. Physica D 388: 45-72.

Bastiaansen R, Jaïbi O, Deblauwe V, Eppinga MB, Siteur K, Siero E, Mermoz S, Bouvet A, Doelman A, Rietkerk M. 2018. Multistability of model and real dryland ecosystems through spatial self-organization. Proceedings of the National Academy of Sciences USA 115:11256-11261.

Nathan J, Osem Y, Shachak M, Meron E. 2016. Linking functional diversity to resource availability and disturbance: a mechanistic approach for water limited plant communities. Journal of Ecology 104: 419-429.

Rietkerk M, Boerlijst MC, van Langevelde F, HilleRisLambers R, van de Koppel J, Kumar L, Prins HHT, De Roos AM. 2002. Self-organization of vegetation in arid ecosystems. American Naturalist 160: 524-530.

Tilman D. 1982. Resource competition and community structure. Princeton University Press, Princeton, NJ, USA.

Comments for the authors:

• Line 17: the term "re-patterning" may read as a non-patterned state becoming patterned again, whereas here it seems to refer to a spatial rearrangement of an existing patterned state.

• Line 39: resources (i.e. plural)?

• Lines 80-99: This is an introduction to the model description, rather than a result, as the header suggests.

• Lines 100-164: This is the model description, which seems to be part of the material and methods rather than a result.

• Line 179: when χ increase from 0.95 to 1.00 however, it seems that the Turing threshold start to increase, how can this reversal be explained?

• Lines 302-310: this explanation is clear, but it is an example that can also be explained by the biomass dynamics of a single variable model.

• Line 329: this is case where it would be useful for readers to understand how one can anticipate the formation of either hybrid or fully patterned states, and how this relates to the particular perturbation(s) imposed.

• Figures: Why are the biomass values in Figure 4,5 and 7 about an order of magnitude higher than in Figure 3?

*Reviewer #2:*

Conspicuous, repetitive patterns such as spots and stripes can be observed in every biome throughout the world. This work provides a new theoretical model for understanding self-organization of vegetation patterns in arid ecosystems and their response to climate (precipitation) change. Processes of spatial self-organization underlying the development of vegetation patterns have been studied for decades, with roots in the work of the great scientist Alan Turing. Ecologists use the Turing reaction-diffusion theory that builds on positive feedback relations between two variables, namely vegetation growth and water transport. Yet, it has been difficult to include multiple, different species as in real-world vegetations.

This paper addresses such shortcoming and extends previous vegetation pattern formation models by including different plant types. It provides a general framework that builds on the resource allocation tradeoff between growth versus stress-tolerance. Authors show when and how vegetation is robust to changes in precipitation via spatial self-organization and selection (differential plant mortality) along the growth-tolerance tradeoff. With increasing aridity, the ecosystem shifts from spatial uniform vegetation to patterned one, such as stripes, and, with further drought, to bare ground. Notably, self-organizing processes mitigate the impact of drought on ecosystem functioning and services by allowing fast-growing, productive species to persist in drier climate. This framework and associated results have important implications for the conservation and management of arid ecosystems and rangelands.

The conclusions of this paper are mostly well supported by data, but some aspects of model presentation, parameter choices, and data interpretation need to be clarified and extended.

1) Model presentation. It would be better to explain the model in ecological terms first, clarifying parameter biological meaning and justifying their choice. In doing so, creating a specific 'Methods' section, which now is lacking, would be of help too. Authors should clarify whether and how the model follows the conservation of mass principle involving precipitation and evapotranspiration. Are root growth and seed dispersal included for this purpose? Why are they not referred to any further in the analysis and discussion? Why a specific term for plant transpiration is not included, or is to somehow phenomenologically incorporated into the growth-tolerance tradeoff? In doing so, authors should also pay attention to water balance as above (H) and below (W) ground water are not independent from each other.

Another unclear point is that growth rates for the same plant functional groups are assumed to be constant among different species within the same group and are confounded by biomass production. Why is that the case? Furthermore, how many different species are characterizing each functional group? How are interspecific interactions accounted for (more specifically, see comment below)?

Finally, stress tolerance is purely phenomenological. There is no actual mechanism/parameter describing it. Rather, it "simply" appears as low/high mortality, which in turn is said to be due to high/low tolerance. This leads to a sort of circularity between mortality and tolerance. Yet, mortality can occur due to other biophysical factors (e.g. disturbance, fire, herbivory, pathogens). A drawback of this assumption is that a mechanism of drought tolerance is often to invest in belowground organs, including roots. However, according to the proposed model, it turns out that fast growing species with low investment in tolerance also have high investment in roots; vice versa, tolerant species have low investment in roots. This is a bit counterintuitive and not well biologically supported.

2) Parameter choice.

N = 128 is an extremely high number for plant functional groups. It is even quite unrealistic to have 128 species per square meter, so this value is not very reasonable. Please run the model and report results with more realistic N (e.g. from 4-64) as well as with different sets of N values keeping all other parameters constant.

Gamma (rate of water uptake by plants' roots): why is it in that unit of m^2^/kg * y? Why are you now considering the area (and not the volume) per biomass unit?

A is not defined in the text.

M min: why 0.5 mortality? Having M max set to 0.9, please consider a lower mortality value set to 0.1, and please report evidence (hopefully) demonstrating the robustness of results to such change.

K_min_ and K_max_ are in two different units, and should both be kg/m^2^.

Values of precipitation (P, mean annual precipitation) are not reported.

3) Results presentation and interpretation.

Parameter range of precipitation in figure 3 is odd. Why in one case precipitation ranges from 0 to 160 while in another it is only 60-120? Furthermore, in paragraph 198-213 and associated results in Figure 5. the Choice of precipitation values is somehow discordant from the previous model. Please provide motivation for this choice, clarify and uniformize it.

Throughout the text, authors claim to address plant-plant interactions, particularly intra and interspecific competition. However, it is not clear how competition was modelled neither whether it was included in the model. In its current state, it is just an assumption pulled out when discussing results – a classic 'passepartout' used by ecologists. Furthermore, why only competition is invoked in interpreting results when facilitation is known to be much more relevant in pattern formation and biodiversity maintenance in arid systems?

Finally, authors seem to create confusion around community composition, which is defined as the (taxonomic) identity of all different species inhabiting a community. Notably, it is remarkably different from the x_max_ parameter used in the model, which as a matter of facts is just the value of the most productive (notably, not necessarily the most abundant) functional group.

*Reviewer #3:*

In this paper, the authors use a mathematical model of plant and water dynamics in drylands to show that drylands adaptive capacity to respond to changes, via spatial self-organization in space has also beneficial effects in preserving its biodiversity and ecosystem functions.

The model is an extension of a large body of previous, well-established works on plant self-organisation in drylands. The model is well described and motivated (with one main exception, see below), the analyses are robust and the results are very convincingly supporting the conclusions. I however have an issue with one of the assumptions in the model equations. The authors included a term for "mutations" in traits that (1) is not introduced or motivated (2) its effects/importance are not highlighted by specific analyses (3) the possible implications or limitations connected to it are not discussed. To my knowledge, this term is also not based on earlier work. All these elements need to be included, as at the moment is for example unclear what the authors intended to represent by including the mutation term (evolutionary time scales? Or adaptation?). Also, it would be especially good to include an analysis of how influential this term is for the final results.

Assuming the authors can address this one concern, the results are surely important as they connect for the first time plant spatial self-organization to its biodiversity preservation, in the face of future expected climatic changes and probable land degradation. These findings, although theoretical, have the potential to be useful also for guiding adaptive and dynamic land management, as they underline the importance of taking into account spatial vegetation distribution in drylands management.

Besides the major point about the mutation term, I list here two other important points:

– The authors state that they represent highly tolerant plants by representing the plants with a small mortality. However, in their model, plant mortality does not depend on soil water levels. How can the authors reconcile these two aspects? Also, one could argue that mortality is related to the average life span, not specifically to tolerance to highly stressful condition. The authors should better justify this point and discuss the implication of this assumption.

– In the model, there is shading feedback too, not only infiltration feedback. However the authors state there's only infiltration feedback in l. 84, could they please explain?

---

## [Author Response]

[Editors’ note: the authors resubmitted a revised version of the paper for consideration. What follows is the authors’ response to the first round of review.]

Reviewer #1:[…]A potential strength of this paper is that the community assembly and biodiversity perspective on spatial self-organization may highlight the relevance of pattern formation in ecosystems more clearly to a broad audience. The formulation of a trait/strategy gradient of discrete classes is certainly an interesting suggestion to connect the typical single/few biomass variable(s) approach to a community-level approach. The community assembly process is modelled in a very specific way, and the manuscript would benefit from an expanded ecological motivation of the processes that are being mimicked, and thereby explain more clearly what taxonomic level of organization is being considered.

We follow the more recent trait-based approach that shifts the focus from species (and the many traits by which they differ from one another) to groups of species that share the same values of selected functional traits. Since the general context is ecosystem response to drier climates, we choose the functional traits to include a response trait associated with stress tolerance and an effect trait associated with biomass production. We further assume a tradeoff between the two traits which is well supported by earlier studies (see e.g. Angert et al. 2009, https://doi.org/10.1073/pnas.0904512106). So, indeed, the choice we make in characterizing the community is quite specific, but it is highly relevant to the ecological context considered of dryland plant communities where plants compete primarily for water and light. The taxonomic level we consider is species except that we group them in a manner that is more transparent to questions of ecosystem function, ignoring differences between species that are not significant to these questions.

We expanded considerably the text in the section “Modeling spatial assembly of dryland plant communities” to clarify the ecological motivation of the processes we model.

In addition, it would be useful if the authors could provide further clarification as to what extent the community diversity dynamics can be separated from total biomass dynamics of patterned water-limited ecosystems given the current approach. These points are explained in further detail below.

The model describes the dynamics of all functional groups, which provides the biomass distribution 𝐵 = 𝐵(𝜒) in trait space (in the case of patterned states we first integrate over space). That distribution contains information about various community-level properties, including functional diversity (richness, evenness) as figure 3 in the revised manuscript illustrates, and total biomass, which is the area below the distribution curve. The two types of dynamics are tightly connected and cannot be separated, but in principle the approach can be used to study the relationships between diversity and total biomass by calculating biomass distributions along the rainfall gradient and extracting the two properties from the distributions.

We added in the section “Modeling spatial assembly of dryland plant communities” the information that the biomass distribution also contains information about the total biomass.

• First, it was not entirely clear to this reviewer how the reaction parts of the model equations determine the optimal trait value χ, and how this value varies as a function of precipitation.

The ‘optimal’ trait value 𝜒_𝑚𝑎𝑥_ is determined by the interspecific interactions that the model captures, which divide into ‘direct’ and ‘indirect’ interactions. The direct interactions are captured by the dependence of the growth rate Λ_𝑖_ of the ith functional group (see Equation 1a) on the aboveground biomass values of all functional groups, Λ_𝑖_ = Λ_𝑖_(𝐵_1_,𝐵_2_,… , 𝐵_𝑁_) (see Equation 2). This dependence represents competition for light (taller plants are better competitors) and includes the effect of self-shading. The indirect interactions are through the water uptake term in the soil-water Equation 1b (2^nd^ term from right) and the water dependence of the biomass growth term in Equation 1a. These terms represent competition for water. For a given precipitation value 𝑃 the net effect of these interspecific interactions result in a particular functional group 𝜒_𝑚𝑎𝑥_ which is most abundant. For spatially uniform vegetation, as 𝑃 is increased 𝜒_𝑚𝑎𝑥_ moves to lower values. The precipitation increases surface water (Equation 1c) and consequently the amount of water 𝐼𝐻 infiltrating into the soil. The increased soil water gives competitive advantage to species investing in growth, mainly because they better compete for light as they grow taller, and therefore 𝜒_𝑚𝑎𝑥_ decreases.

Assuming a single trait class, and plotting the relevant equilibrium values of the three state variables shed some light on this issue. [Unfortunately, there does not seem to be a possibility to attach the figure with these plots to this review report]. Assuming the non-spatial equilibrium solution was derived correctly , the optimum biomass (B) value shifts across the trait spectrum with changing precipitation (in the non-spatial model version, solving the surface water equation for equilibrium will always yield that all precipitation infiltrates, i.e. regardless of the values of surface water, H, and χ). The equilibrium of soil water availability (W), which is the growth limiting resource of the vegetation, shows an inverse pattern with biomass. This result is in line with a classical results (e.g. Tilman 1982), in that the most successful strategy is the one that is able to reduce the limiting resource to the lowest equilibrium value. With all trait classes competing for the soil water resource, however, it is then not immediately clear why the most successful trait class is not outcompeting the other classes. This leads to a second point, about the way in which community trait adaptation is modelled.

With the current model and parameters set the most successful trait does eventually outcompete all other traits, when trait diffusion is set to zero, 𝐷_𝜒_ = 0. This is, however, a very long process because the most successful trait suffers from self-shading at late growth stages, which slows down its growth and allows nearby traits to survive for a long time. Choosing a finite but very small 𝐷_𝜒_ values that represent mutations occurring on evolutionarily long times counteracts the exclusion process and results in a stationary asymptotic community, as Figure 3 in the revised manuscript shows (this behavior is reminiscent of optical solitons, where self-focusing instability is balanced by dispersion). We note that modeling stronger growth-inhibiting factors, such as pathogens, by including a factor of the form (1 − 𝐵_𝑖_/𝐾) to the growth rate, results in an asymptotic stationary community also for 𝐷_𝜒_ = 0 (see also earlier studies Nathan et al. 2016, Yizhaq et al. 2020).

We revised original Figure 4 (now Figure 3) by adding a new part (Figure 3a) that shows the exclusion process for 𝐷_𝜒_ = 0, and the effect of the counter-acting process of trait diffusion, which results in an asymptotic distribution of finite width (Figure 3b) from which community level properties such as functional diversity can be derived. We also extended the text in section “Modeling spatial assembly of dryland plant communities” (last paragraph) to clarify the two counter-acting processes of exclusion because of interspecific competition for water and light, and trait diffusion driven by mutations, which together culminate in an asymptotic biomass distribution along the 𝜒 axis of finite width.

• The authors model trait adaptation through a diffusion approximation between trait classes. That is, every timestep, a small amount of biomass flows from the class with higher biomass to the neighboring trait class with lower biomass. From an ecological point of view, it seems that this process is describing adaptation of vegetation that is already present, so this process seems to be limited to intraspecific phenotypic plasticity. From the text, however, it seems that the trait classes correspond to higher taxonomic levels of organization, when describing shifts from fast growing to stress-tolerant species, for example. It is not entirely clear, however, how biomass flows as assumed in the model could occur at these higher levels of organization.

We do not study in this work adaptation through diffusion in trait space. That kind of adaptive dynamics can indeed be studied with the current model, but with different initial conditions, namely, initial conditions corresponding to a single resident trait where the biomass of all other traits is zero. The resulting dynamics of mutations and succession are then very slow, occurring on evolutionarily long time scales set by the small value of 𝐷_𝜒_ (e.g. 10^−6^). In this study the initial conditions represent the presence of all traits, even if at very low biomass values that may represent a pool of seeds that germinate once environmental conditions allow. For a given precipitation value 𝑃, the functional traits we consider determine which functional groups (of species) overcome environmental filtering and grow, and which of the growing traits survive the competition for water and light. These are relatively fast processes, occurring on ecological time scales, which determine the emerging community. At longer times this community is further shaped by slow processes of interspecific competition among species of similar traits and by trait diffusion (mutations). A final remark about phenotypic changes: although in general 𝜒 can be interpreted as representing different phenotypes, the choice of very small values for 𝐷_𝜒_ cannot represent relatively fast phenotypic changes and restricts the context to mutations at the taxonomic level of species.

We added an explanation in the 3^rd^ paragraph of the section “Modeling spatial assembly of dryland plant communities” of the need to consider mutations and the role they play in our study.

• Combining the observations from the previous two points, there is a concern that for a given level of precipitation, there is a single trait class with optimal biomass/lowest soil water level that is dominant, with the neighboring trait classes being sustained by the diffusion of biomass from the optimal class to neighboring inferior classes. This would seem a bit problematic, as it would mean that most classes are not a true fit for the environment, and only persist due to the continuous inflow of biomass. Taking a clue from the previous papers of the authors, it seems this may not be the case, though. Specifically, in the paper by Nathan et al. (2016) it seems that all trait classes are started at low initial biomass density, and the resulting steady state (in the absence of biomass flows between classes) seems to show similar biomass profiles as shown in Figures 4,5 and 7 of the current paper. While the current model formulation seems slightly different, similar results may apply here. Indeed, keeping all trait classes at non-zero (but low) density, and when the (abiotic and biotic) environment permits, let each class increase in biomass seems like the most straightforward approach to model community assembly dynamics. Given the above discussion about these trait classes competing for a single resource (soil water), and one trait class being able to drive this resource availability to the lowest level, it would then be useful to readers to explain why multiple trait classes can coexist here, and how (for spatial uniform solutions) the equilibrium soil water level with multiple trait classes present compares to the equilibrium soil water level when only the optimal trait class is present. Furthermore, if results as presented in Nathan et al. (2016) indeed hold in the current case, perhaps it means that the biomass profile responses as shown in e.g. Figure 5 would also occur if there was no biomass flow between trait classes included, but that the time needed to adjust the profile would take much longer as compared to when the drift term/second trait derivative is included. In summary, further clarification of what the biomass flows between classes represent, and the role it plays in driving the presented results would be useful for readers.

As explained in the reply to previous comments the asymptotic community is tuned by a balance between two slow counter-acting processes, interspecific competition among similar traits and mutations over evolutionarily long time scales. However, the community structure is largely determined by much faster processes of environmental filtering and interspecific competition among widely distinct traits, as all traits are initially present. Indeed, comparing the biomass distributions in new Figure 3, with and without trait diffusion indicates that the community composition, as measured by 𝜒_𝑚𝑎𝑥_, is the same. Trait diffusion, however, does affect functional diversity, along with environmental factors. In that sense the emerging community is a true fit for the environment.

We thank the reviewer for these thoughtful comments, which helped us realize that our presentation of these issues was too concise and unclear. We believe that the new extended section on modeling spatial assembly of dryland plant communities, and the new figure 3a clarify these issues.

• In addition, it would be useful for readers to understand to what extent the shifts in average trait values and functional diversity can be decoupled from the biomass and soil water responses to changes in precipitation that would occur in a model with only a single biomass variable. For example, early studies on self-organization in semi-arid ecosystems already showed that the shift toward a patterned state involved the formation of patches with higher biomass, and higher soil water availability, as compared to the preceding spatially uniform state, and that the biomass in these patches remains relatively stable under decreasing rainfall, while their geometry changes (e.g. Rietkerk et al. 2002). It has also been observed that for a given environmental condition, biomass in vegetation patches tends to increase with pattern wavelength (e.g. Bastiaansen and Doelman 2018; Bastiaansen et al. 2018). Given the model formulation, one wonders whether higher biomass in the single variable model is not automatically corresponding to higher abundance of faster growing species and a higher functional diversity (as the diffusion of biomass can cover a broader range when starting from higher mass in the optimal trait class). There are some indications in the current work that the linkage is more complicated, for example, the biomass peak in Figure 7c is lower, but also broader as compared to the distribution of Figure 7b, but it is currently not entirely clear how this result can be explained (for example, it might be the case that in the spatially patterned states, the biomass profiles also vary in space).

We are not sure we understand what the reviewer means by “decoupled”, but much insight indeed can be gained from a study of a model for a single functional group (trait) and observing the behaviors described by the reviewer. In fact, these behaviors, which some of us are familiar with from numerical studies, motivated parts of the current study. Higher biomass in vegetation patches (compared to uniform vegetation) in the single trait model does not automatically imply a shift to faster growing species; in principle the stress-tolerant species that already reside in the system when uniform vegetation destabilizes to a periodic pattern can simply grow denser. To answer this and additional questions we need to take into account interspecific interactions by studying the full community model. As to Figure 7b,c, the behavior appears to be opposite to that described by the reviewer: the biomass pick in Figure 7c is higher and narrower than that in Figure 7b, not lower and broader. This is because of the much larger domain of the patterned state as compared with that of the uniform state, which increases the abundance of low-𝜒 species, i.e. species investing in growth.

The increase of biomass in vegetation patches with pattern wavelength for given environmental conditions, as observed by Bastiaansen et al. 2018, is actually another mechanism for increasing functional diversity. This is because the water stress at the patch center is higher than that in the outer patch areas and thus forms favorable conditions for stress tolerant species while the outer areas form favorable conditions for fast growing species.

We added a new paragraph in the Discussion and Conclusion section (last paragraph in the subsection Insight III) where we discuss the effect of coexisting periodic patterns of different wavelengths on functional diversity and ecosystem management. We also added citations to the references the reviewer mentioned.

• The possibility of hybrid states, where part of the landscape is in a spatially uniform state, while the other part of the landscape is in a patterned state, is quite interesting. To better understand how such states could be leveraged in management strategies, it would be useful if a bit more information could be provided on how these hybrid states emerge, and whether one can anticipate whether a perturbation will grow until a fully patterned state, or whether the expansion will halt at some point, yielding the hybrid state. It seems that being able to distinguish these case would be necessary in the design of planning and management strategies.

The hybrid states appear in the bistability range of the uniform and patterned vegetation states, and typically occupy most of this range. Their appearance is related to the behavior of ‘front pinning’ in bistability ranges of uniform and patterned states in general. Front pinning refers to fronts that separate a uniform domain and a periodic-pattern domain, which remain stationary in a *range* of a control parameter (precipitation in our case). This is unlike fronts that separate two uniform states, which always propagate in one direction or another and can be stationary only at a single parameter value – the Maxwell point. Thus, an indication that a given landscape may have the whole multitude of hybrid states is the presence of a front (ecotones) that separates uniform and patterned vegetation. If that front appears stationary over long period of times (on average), this is a strong indication.

We added a new paragraph in the subsection Insight III of the Discussion and conclusion section to clarify this point.

Also, in Figure 3a, the region of parameter space in which hybrid states occur is not very large; it is not entirely clear whether the full range of hybrid states is left out here for visual considerations, or whether these states only occur within this narrow range in the vicinity of the Turing instability point.

As pointed out in the reply to the previous comment the hybrid states are limited to the bistability range of uniform and patterned vegetation, which is not wide. However, this should not necessarily restrictma nagement of ecosystem services by nonuniform biomass removal, as such management will have similar effects on community structure also outside the bistability range where front propagate slowly.

The new paragraph we added also addresses this point.

Comments for the authors:• Line 17: the term "re-patterning" may read as a non-patterned state becoming patterned again, whereas here it seems to refer to a spatial rearrangement of an existing patterned state.

We change “spatial re-patterning” to “spatial self-organization”

• Line 39: resources (i.e. plural)?

Fixed.

• Lines 80-99: This is an introduction to the model description, rather than a result, as the header suggests.• Lines 100-164: This is the model description, which seems to be part of the material and methods rather than a result.

We added a new section “Methods” and moved the more technical part of the model description to that section.

• Line 179: when χ increase from 0.95 to 1.00 however, it seems that the Turing threshold start to increase, how can this reversal be explained?

This is a good question, which we are not sure we have a definite answer to, because the analytical expression for the Turing threshold is too complicated to be helpful in this regard. A possible explanation, however, is the following. Spatial patterning in this model is driven by a high infiltration contrast between vegetation patches and bare soil, as this increases the flow speed of overland water. Thus, high infiltration contrast (obtained for patches of high biomass density) favors higher thresholds 𝑃_𝑇_. This explains why low 𝜒 species, which invest in growth, have higher 𝑃_𝑇_. However, the shift to higher 𝜒 species at low P not only implies less investment in growth and thus lower biomass density, but also weaker water uptake which leaves more soil-water content for growth. In any event the non-monotonic behavior of 𝑃_𝑇_ at the high 𝜒 range does not affect the results we present as these are all related to behaviors occurring for 𝜒 values well below that range.

• Lines 302-310: this explanation is clear, but it is an example that can also be explained by the biomass dynamics of a single variable model.

This explanation relies indeed on the dynamics of a single functional-group model and is used to argue that similar results about community dynamics should be obtained in 2d where morphological transitions can take place, a case we do not study in this work.

• Line 329: this is case where it would be useful for readers to understand how one can anticipate the formation of either hybrid or fully patterned states, and how this relates to the particular perturbation(s) imposed.

This point is discussed in a new paragraph we added in this section (paragraph next to last).

• Figures: Why are the biomass values in Figure 4,5 and 7 about an order of magnitude higher than in Figure 3?

We thank the reviewer for raising this point. In fact the biomass values in Figure 4 (now 3), 5 and 7 should be an order of magnitude *lower* than in in Figure 3 (now 4). The reason is that there are about 30 functional groups in the community that share the water content and therefore the biomass of each functional group should be much smaller compared to the case of a single functional group that benefits from the whole amount of water (Figure 3 in original manuscript). This confusion is a consequence of a different definition of the biomass variables used in the community model vs. that used in the single functional-group model. A biomass variable 𝐵_𝑖_ in Equation 1 represents biomass density in trait space, which we now denote as 𝑏_𝑖_, rather than the biomass of the *i*^th^ functional group, which is 𝐵_𝑖_ = 𝑏_i_Δ𝜒 = 𝑏_𝑖_/𝑁, where 𝑁=128 in our study.

We rewrote the community model – Equation 1 (appearing now in the new Methods section) in terms of the new biomass variables 𝐵_𝑖_ = (𝑏_i_Δ𝜒) and updated the vertical biomass axes in Figures 4 (now 3), 5 and 7. We also added an explanation that the single functional-group model with 𝜒 = 1 used to produce Figure 3 (now 4) follows from the community model by setting all biomass variables identically to zero apart of 𝐵_𝑁_ which corresponds to 𝜒_𝑁_ = 𝑁Δ𝜒 = 1 (and dropping the trait-diffusion term which does not have a meaning for a single functional-group model).

Reviewer #2:[…]The conclusions of this paper are mostly well supported by data, but some aspects of model presentation, parameter choices, and data interpretation need to be clarified and extended.1) Model presentation. It would be better to explain the model in ecological terms first, clarifying parameter biological meaning and justifying their choice. In doing so, creating a specific 'Methods' section, which now is lacking, would be of help too. Authors should clarify whether and how the model follows the conservation of mass principle involving precipitation and evapotranspiration. Are root growth and seed dispersal included for this purpose? Why are they not referred to any further in the analysis and discussion? Why a specific term for plant transpiration is not included, or is to somehow phenomenologically incorporated into the growth-tolerance tradeoff? In doing so, authors should also pay attention to water balance as above (H) and below (W) ground water are not independent from each other.

We added a Methods section, which in *eLife* is placed at the end of the manuscript. The section includes the model equations and more detailed explanations in ecological terms of various parts of the model. We also added Table 1 with a list of all model parameters, their descriptions, units and numerical values used in the simulations. Presenting the model at the end of the manuscript suits more technical information about the model, but not essential information that is needed for understanding the results. We therefore kept the subsection “A model for spatial assembly of dryland plant communities” in the Results section, where we present that information.

There is no conservation of mass in the model (and all other models of this kind) simply because the system that we consider is open. In particular, it does not include the atmosphere, which constitute part of the system’s environment. Including the atmosphere as additional state variables in the model, capturing the feedback of evapotranspiration on the atmosphere, would make the model too complicated for the kind of analysis we perform. So, although the model contains parts that represent mass conservation such as the terms describing below- and above-ground water transport, water mass is not conserved. The biomass variables represent aboveground biomass of living plants or plant parts and are not conserved either as biomass production involve biochemical reactions that convert inorganic substances coming from the system’s environment (atmosphere and the soil) into organic ones, while plant mortality involves organic matter that leaves the system.

Roots in the model platform we consider are modeled indirectly through their relation to aboveground biomass. That relation constitutes one of the scale-dependent feedbacks that produce a Turing instability to vegetation patterns, the so-called root-augmentation feedback (see Meron 2019, Physics Today), but in this particular study we eliminate this feedback for simplicity. The scale-dependent feedback that we do consider is the so-called infiltration feedback, associated with biomass-dependent infiltration rate that produces overland water flow towards vegetation patches, as explained in the subsection “A model for spatial assembly of dryland plant communities”. It will be interesting indeed to extend the study in the future to include also the root-augmentation feedback.

We assume short-range seed dispersal and take it into account through biomass “diffusion” terms (obtained as approximations of dispersal kernels assuming narrow kernels). These terms play important roles in the scale-dependent feedback that induces the Turing instability, as is explained in earlier papers which we cite. Plant transpiration is modeled through the water uptake term in the equation for the soilwater 𝑊. Indeed above-ground water 𝐻 and below-ground water 𝑊 are not independent; the infiltration term IH in the equations for both state variables account for this dependence in a unidirectional manner (loss of 𝐻 and gain of 𝑊). As we do not include the atmosphere in the model the other direction, namely, evapotranspiration that increases air humidity and affects rainfall, is not accounted for. The neglect of this effect can be justified for sparse dryland vegetation.

These good points have already been discussed in many earlier papers as well as in the book Nonlinear Physics of Ecosystems (Meron 2015), and we cannot address them all in this paper. We did however add several clarifications in the section Modeling spatial assembly of dryland plant communities and in the new Methods section, including the consideration of the atmosphere as the system’s environment quantified by the precipitation parameter 𝑃.

Another unclear point is that growth rates for the same plant functional groups are assumed to be constant among different species within the same group and are confounded by biomass production. Why is that the case? Furthermore, how many different species are characterizing each functional group? How are interspecific interactions accounted for (more specifically, see comment below)?

In the trait-based approach we focus on just two functional traits, related to growth rate and tolerance to water stress, ignoring differences in other traits that distinguish species. That is, a given functional group consists of species that share the same values of the two selected functional traits (to a given precision determined by 𝑁), taking all other traits represented in the model to be equal. In this approach we do not care about how many species belong to each functional group, only their total biomass. We wish to add that simplifying assumptions of this kind are necessary if we want the model to be mathematically tractable and capable of providing deep insights by mathematical analysis.

We expanded the discussion of the trait-based approach in the section Modeling spatial assembly of dryland plant communities and added relevant references (second paragraph).

Finally, stress tolerance is purely phenomenological. There is no actual mechanism/parameter describing it. Rather, it "simply" appears as low/high mortality, which in turn is said to be due to high/low tolerance. This leads to a sort of circularity between mortality and tolerance. Yet, mortality can occur due to other biophysical factors (e.g. disturbance, fire, herbivory, pathogens). A drawback of this assumption is that a mechanism of drought tolerance is often to invest in belowground organs, including roots. However, according to the proposed model, it turns out that fast growing species with low investment in tolerance also have high investment in roots; viceversa, tolerant species have low investment in roots. This is a bit counterintuitive and not well biologically supported.

First, we agree with the reviewer that our approach is purely phenomenological, as we model tolerance to water stress by a single parameter that lumps together the effects of various physiological mechanisms. That parameter can be distinguished from other factors affecting mortality by regarding the constant 𝑀_𝑚𝑎𝑥_ in Equation 3 as representing several contributions. Since we do not study the effects of these other factors we can absorb them in 𝑀_𝑚𝑎𝑥_ for mathematical simplicity.

Tolerance to water stress is not necessarily associated with roots. Plants can better tolerate water stress by reducing transpiration through stomatal closure, regulating leaf water potential, or develop hydraulically independent multiple stems that lead to a redundancy of independent conduits and higher resistance to drought (see Schenk et al. 2008 – https://doi.org/10.1073/pnas.0804294105).

We added a discussion in the Methods section (5^th^ paragraph, “Tolerance to water stress …”) of the simple form by which we model tolerance to water stress through the mortality parameter.

2) Parameter choice.N = 128 is an extremely high number for plant functional groups. It is even quite unrealistic to have 128 species per square meter, so this value is not very reasonable. Please run the model and report results with more realistic N (e.g. from 4-64) as well as with different sets of N values keeping all other parameters constant.

We wish to clarify two points: (1) N=128 does not imply 128 functional groups per square meter; the emerging community has much lower functional richness (FR) as the average FR is around 0.25, meaning only 128 × 0.25 = 32 functional groups. (2) The model results, as reflected by the key metrics 𝜒_𝑚𝑎𝑥_, 𝐹𝑅, and 𝐹𝐸, are *independent* of the particular value of N (for N values sufficiently large), as Author response image 1 show. The biomass 𝐵_𝑖_ of each functional group, however, does change (Author response image 1) because by changing N we change the range of traits Δ𝜒 = 1/𝑁 that belong to a given functional group. But if we look at the biomass density in trait space 𝑏_𝑖_, related to 𝐵_𝑖_ through the relation 𝐵_𝑖_ = 𝑏_𝑖_Δ𝜒, then also the biomass density is independent of 𝑁 as Author response image 1 shows. So, even if in practice there are less functional groups and thus species as considered in the model studies, the results are not affected by that. On the other hand, choosing higher 𝑁 values provides smoother curves and nicer presentation of our results.

We added a discussion of this issue in the Methods section after Equation 2.

Gamma (rate of water uptake by plants' roots): why is it in that unit of m^2^/kg * y? Why are you now considering the area (and not the volume) per biomass unit?

The vegetation pattern formation model we study, like most other models of this kind, does not explicitly capture the soil depth dimension. Accordingly, W is interpreted as the soil-water content in the soil volume below a unit ground area within the reach of the plant roots. In practice W has units kg/m^2^, like B, and since Γ𝑊𝐵 should have the same units as 𝜕𝑊/𝜕𝑡 see Equation 1b, Γ must have the units of (𝐵𝑡)^−1^.

A is not defined in the text.

We now define it in Table 1 (see Methods section).

M min: why 0.5 mortality? Having M max set to 0.9, please consider a lower mortality value set to 0.1, and please report evidence (hopefully) demonstrating the robustness of results to such change.

The results are robust to the particular values of 𝑀_𝑚𝑖𝑛_ and 𝑀_𝑚𝑎𝑥_, except that there are combinations of these two parameters for which the biomass distributions are pushed towards the edge of the 𝜒 domain, which make the presentation of the results less clear. Author response image 2 shows results of recalculations of the distribution 𝐵 = 𝐵(𝜒) for 𝑀_𝑚𝑖𝑛_ = 0.1, as requested (using 𝑀_𝑚𝑎𝑥_ = 0.15) for 3 different precipitation values. As the reviewer can see there’s no qualitative change in the results: lower precipitation push a uniform community to stress tolerant species (higher 𝜒), while the formation of patterns at yet lower precipitation push the community back to fast growing species (low 𝜒).

**Author response image 2. respfig2:** 

K_min_ and K_max_ are in two different units, and should both be kg/m^2^.

Thanks, we fixed this typo in Table 1.

Values of precipitation (P, mean annual precipitation) are not reported.

The precipitation parameter is variable, as is now stated in Table 1, and therefore was not include it in the list of parameters’ values used. Whenever a particular precipitation value has been used our intention was to state it in the caption of the corresponding figure. This was done in Figures 5,6,7, but indeed not in Figure 4 (Figure 3 in revised manuscript). The insets on the right side of Figure 3 (Figure 4 in revised manuscript) where also calculated for particular precipitation values, but that information is not essential as the intention is to show *typical* forms of the various solution branches, which do not qualitatively change along the branches (i.e. at different P values).

We added the precipitation value (P=180mm/y) at which all the biomass distributions shown in new Figure 3 (Figure 4 in original manuscript) were calculated.

3) Results presentation and interpretation.Parameter range of precipitation in figure 3 is odd. Why in one case precipitation ranges from 0 to 160 while in another it is only 60-120? Furthermore, in paragraph 198-213 and associated results in Figure 5. the Choice of precipitation values is somehow discordant from the previous model. Please provide motivation for this choice, clarify and uniformize it.

In Figure 3b (Figure 4b in revised manuscript) we restricted the precipitation range to 60-120 as the curves, which are limited to 0 < 𝜒 < 1 (by the definition of 𝜒), do not extend to 𝑃 < 60 and to 𝑃 > 120. Extending the range to 0 < 𝑃 < 160 would make the figure less compact and nice as it will contain blank parts with no information.

We are not sure we understand what the reviewer means by “is somehow discordant from the previous model”. The motivation of the choices we made for the precipitation values P=150, 100 and 80 was to show the shift of a spatially uniform community to a higher 𝜒 value as the precipitation is decreased to a lower value (from 150 to 100), and the shift back to a lower 𝜒 value at yet lower precipitation (80) past the Turing instability.

Throughout the text, authors claim to address plant-plant interactions, particularly intra and interspecific competition. However, it is not clear how competition was modelled neither whether it was included in the model. In its current state, it is just an assumption pulled out when discussing results – a classic 'passepartout' used by ecologists. Furthermore, why only competition is invoked in interpreting results when facilitation is known to be much more relevant in pattern formation and biodiversity maintenance in arid systems?Finally, authors seem to create confusion around community composition, which is defined as the (taxonomic) identity of all different species inhabiting a community. Notably, it is remarkably different from the x_max_ parameter used in the model, which as a matter of facts is just the value of the most productive (notably, not necessarily the most abundant) functional group.

We thank the reviewer for this comment. Since all the emerging communities in the model studies are pretty localized around the value of 𝜒_𝑚𝑎𝑥_, that value does contain information about the identity of other functional groups in the community when complemented by FR (functional richness) and FE (functional evenness). More significantly to our study, shifts in 𝜒_𝑚𝑎𝑥_ represent the shifts in community composition we focus on in this study, i.e. shifts towards fast growing species or towards stress-tolerant species.

We modified the description of the community-level properties that can be derived from the biomass distribution in trait space (see modified text towards the end of the section “Modeling spatial assembly …” and also the caption of Figure 3b), explaining that both functional diversity and community composition can be described by several metrics, and clarifying the significance of 𝜒_𝑚𝑎𝑥_ in describing community composition shifts.

Reviewer #3:In this paper, the authors use a mathematical model of plant and water dynamics in drylands to show that drylands adaptive capacity to respond to changes, via spatial self-organization in space has also beneficial effects in preserving its biodiversity and ecosystem functions.The model is an extension of a large body of previous, well-established works on plant self-organisation in drylands. The model is well described and motivated (with one main exception, see below), the analyses are robust and the results are very convincingly supporting the conclusions. I however have an issue with one of the assumptions in the model equations. The authors included a term for "mutations" in traits that (1) is not introduced or motivated (2) its effects/importance are not highlighted by specific analyses (3) the possible implications or limitations connected to it are not discussed. To my knowledge, this term is also not based on earlier work. All these elements need to be included, as at the moment is for example unclear what the authors intended to represent by including the mutation term (evolutionary time scales? Or adaptation?). Also, it would be especially good to include an analysis of how influential this term is for the final results.Assuming the authors can address this one concern, the results are surely important as they connect for the first time plant spatial self-organization to its biodiversity preservation, in the face of future expected climatic changes and probable land degradation. These findings, although theoretical, have the potential to be useful also for guiding adaptive and dynamic land management, as they underline the importance of taking into account spatial vegetation distribution in drylands management.Besides the major point about the mutation term, I list here two other important points:

We addressed this issue in our reply to the first reviewer and summarize our reply as follows. Without the mutation term interspecific competition among all functional groups (assumed to be present initially) results in the emergence of a community centered around a most abundant functional group 𝜒_𝑚𝑎𝑥_ on a relatively fast ecological time scale. This functional group eventually outcompetes all other functional groups, but that competition becomes extremely slow as the competing functional groups become similar. On these long-time scales mutations is a process that cannot be ignored. Including mutations at evolutionarily slow rates (𝐷_𝜒_ = 10^−6^) results in the emergence of an asymptotic community with a characteristic distribution of traits (functional diversity) around the same 𝜒_𝑚𝑎𝑥_. These results are also displayed graphically in new Figure 3.

– The authors state that they represent highly tolerant plants by representing the plants with a small mortality. However, in their model, plant mortality does not depend on soil water levels. How can the authors reconcile these two aspects? Also, one could argue that mortality is related to the average life span, not specifically to tolerance to highly stressful condition. The authors should better justify this point and discuss the implication of this assumption.

The relation to the soil-water level is as follows. The fitness of the ith functional group is Λ_𝑖_𝑊 − 𝑀_𝑖_. This means that functional groups with lower mortality can survive (i.e. have positive fitness) lower soil-water contents or higher water stress. This simple modeling form applies to situations where stress tolerance is not a phenotypic trait switched on by water stress. An example would be plants with hydraulically independent multiple stems that lead to a redundancy of independent conduits and higher resistance to drought (see Schenk et al. 2008 - https://doi.org/10.1073/pnas.0804294105). We could model stress tolerance as a phenotypic trait, as was done in Tzuk et al. 2019, but that would not change the main results and message. There are obviously additional factors contributing to plant mortality besides water stress. However, since we do not study these factors in this work we could lump their contributions in the mortality parameter 𝑀_𝑚𝑎𝑥_.

We added a discussion of these issues in the (new) Methods section (paragraph beginning with “Tolerance to water stress is modeled …”).

– In the model, there is shading feedback too, not only infiltration feedback. However the authors state there's only infiltration feedback in l. 84, could they please explain?

The shading feedback (reduced evaporation by shading) is not a scale-dependent feedback that leads to spatial patterning, as it does not involve water transport. It rather leads to bistability of states.